# Learning to Specialize: Joint Gating-Expert Training for Adaptive MoEs in Decentralized Settings

**Yehya Farhat**[*][†]     **Hamza ElMokhtar Shili**[*][†]     **Fangshuo Liao**[*][†]     **Chen Dun**[*][†]

**Mirian Del Carmen Hipolito Garcia**[*][‡]          **Guoqing Zheng**[§]

**Ahmed Hassan Awadallah**[‡]          **Robert Sim**[‡]          **Dimitrios Dimitriadis**[§]

**Anastasios Kyrillidis**[†]
Department of Computer Science, Rice University
Microsoft

## Abstract

Mixture-of-Experts (MoEs) achieve scalability by dynamically activating subsets of their components. Yet, understanding how expertise emerges through joint training of gating mechanisms and experts remains incomplete, especially in scenarios without clear task partitions. Motivated by inference costs and data heterogeneity, we study how joint training of gating functions and experts can dynamically allocate domain-specific expertise across multiple underlying data distributions. As an outcome of our framework, we develop an instance tailored specifically to decentralized training scenarios, introducing *Dynamically Decentralized Orchestration of MoEs* or DDOME. DDOME leverages heterogeneity emerging from distributional shifts across decentralized data sources to specialize experts dynamically. By integrating a pretrained common expert to inform a gating function, DDOME achieves personalized expert subset selection on-the-fly, facilitating just-in-time personalization. We empirically validate DDOME within a Federated Learning (FL) context: DDOME attains from 4% up to a 24% accuracy improvement over state-of-the-art FL baselines in image and text classification tasks, while maintaining competitive zero-shot generalization capabilities. Furthermore, we provide theoretical insights confirming that the joint gating-experts training is critical for achieving meaningful expert specialization.

## 1 Introduction

Due to the success of large-scale deep learning [28, 2, 3, 15, 5, 18], it is now widely accepted as a design philosophy that "*the larger (model/dataset), the better*". Yet, the computational and economic costs associated with training and deploying large monolithic models raise concerns: *Are we wisely allocating resources by training a single, monolithic model rather than employing specialized submodels that operate efficiently and adaptively?*

Mixture of experts (or MoEs) [10, 12] represent a well-known model architecture that embodies this principle. Unlike traditional ensemble methods [7], MoEs train expert submodels jointly, using

---

[*]Equal contribution.
[†]{yehya.farhat, hamza.shili, fangshuo.liao, cd46, anastasios}@rice.edu
[‡]{mirianh, ahmed.awadallah, rsim}@microsoft.com
[§]Work done while at Microsoft.{gzheng}@percepta.ai, {dbdim}@amazon.com

39th Conference on Neural Information Processing Systems (NeurIPS 2025).

a gating function to selectively activate subsets of experts based on input data. Recent successful instances of MoEs demonstrate impressive scalability benefits by leveraging sparse activation of network layers [30, 5, 18, 27, 25, 32].

Despite these successes, achieving *semantic specialization* –where experts learn distinct functions– remains challenging in single-domain tasks, where data does not naturally partition into subtasks or distinct distributions. Recent findings even suggest that sophisticated gating strategies might offer limited benefit compared to random gating functions in such single-domain contexts [36], indicating that explicit specialization does not emerge without additional mechanisms or constraints. Existing methods [19, 6] addressing meaningful specialization primarily focus on distinct task boundaries and separate datasets per expert to facilitate explicit expert specialization. This observation leads to a fundamental question: *How can we effectively foster meaningful specialization within MoEs in single-domain scenarios, where clear task boundaries are not naturally available?*

Understanding this question has broad implications for real-world settings where data exhibits subtle variations rather than clear domain boundaries. For instance, consider pretraining a model to predict the type and stage of cancer from CT scans. In this case, cancer prediction is the shared task, while the clients—different hospitals or clinics—each possess private datasets reflecting their own data distributions (e.g., one may specialize in prostate cancer, another in breast cancer). Similar situations arise in personalized content recommendation, language modeling across dialects, and client-specific adaptation in decentralized environments. Studying this phenomenon can lead to strategies for dynamically allocating computational resources, thereby improving both model efficiency and overall performance.

**Main hypothesis and our contributions.** We explore how joint training of gating mechanisms and experts within MoE frameworks can enable adaptive specialization under single-domain settings. We hypothesize that MoEs can leverage implicit data heterogeneity to achieve expert specialization, guided by a dynamic gating function that learns concurrently with experts.

As an instance, we introduce a decentralized MoE framework designed to leverage data characteristics across different nodes for personalized expert specialization. We dub this system *Dynamically Decentralized Orchestration of MoEs* or DDOME, a distributed MoE system tailored for decentralized learning scenarios (Figure 1). DDOME maintains a collection of independent expert models (we consider image and text classification tasks in this work), adaptively selected by a gating function influenced by shared representations from a pretrained common expert. This design allows DDOME to dynamically specialize subsets of experts across heterogeneous data distributions without explicit task annotations. Some of our findings include:

- We theoretically show decoupling the training of the gating and expert modules leads to suboptimal specialization, highlighting the necessity of joint training for effective expert allocation.
- DDOME effectively leverages implicit client-specific data characteristics to dynamically specialize experts during joint training, without the need for explicit task definitions.
- DDOME can dynamically select experts and achieve just-in-time personalization on unseen clients during testing. DDOME accurately classify unseen data with small adaptations.
- We achieve these reducing the overall communication cost by not sending the whole MoE module to all clients, compared to state-of-the-art methods [29].
- Some highlights of DDOME in practice: Within a Federated Learning (FL) setup, DDOME achieves $\sim 95\%$ accuracy on FL CIFAR10, $\sim 78\%$ accuracy on FL CIFAR100, and $\sim 75\%$ on FL Yahoo! Answers text classification as a *just-in-time personalization* method on unseen clients, where the second best SOTA method achieves $\sim 71\%$, $\sim 74\%$, and $\sim 69\%$ respectively.

## 2 Background

**Notation.** Vectors and matrices are represented with bold font (e.g., $\mathbf{x}$), while scalars by plain font (e.g., $x$ or $S$). Capital letters distinguish matrices from vectors (e.g., $\mathbf{W}$ vs $\mathbf{w}$). Calligraphic uppercase letters denote sets (e.g., $\mathcal{D}$); the cardinality of $\mathcal{D}$ is represented as $|\mathcal{D}|$. $[N]$ is $[N] = \{1 \dots N\}$.

**Problem formulation.** Let $S$ be the total number of training clients. Each client $s$ has its own local data, denoted as $\mathcal{D}_s$. We will assume that $\mathcal{D}_s = \{\mathbf{x}_i, y_i\}_{i=1}^{|\mathcal{D}_s|}$, where $\mathbf{x}_i$ is the $i$-th input sample and $y_i$ its corresponding label in a supervised setting. Abstractly, let $\mathbf{W}$ denote the collection of trainable model parameters. The goal is to find values for $\mathbf{W}$ that achieve good accuracy on all data $\mathcal{D} = \cup_s \mathcal{D}_s$, by minimizing the following optimization objective:

$$\mathbf{W}^\star \in \arg\min_{\mathbf{W}} \left\{ \mathcal{L}(\mathbf{W}) := \tfrac{1}{S} \sum_{s=1}^{S} \ell(\mathbf{W}, \mathcal{D}_s) \right\},$$

where $\ell(\mathbf{W}, \mathcal{D}_s) = \frac{1}{|\mathcal{D}_s|} \sum_{\{\mathbf{x}_i, y_i\} \in \mathcal{D}_s} \ell(\mathbf{W}, \{\mathbf{x}_i, y_i\})$. Here, with a slight abuse of notation, $\ell(\mathbf{W}, \mathcal{D}_s)$ denotes the *local* loss function for user $s$, associated with a local model $\mathbf{W}_s$ (not indicated above), that gets aggregated with the models of other users. $\mathbf{W}_s$ could be a full copy of the global model at the current training round or a selected submodel out of the global one, randomly chosen or based on the client's characteristics.

It is desired that the trained global model $\widehat{\mathbf{W}} \approx \mathbf{W}^\star$ is applied to unseen test clients that come with different non-i.i.d local data. Previous approaches handling a similar scenario [33, 31] assume we have access to part of the new client's labeled local data and fine-tune $\widehat{\mathbf{W}}$. We consider this a limitation since new users are likely unwilling/unable to provide accurate labeled data and might not have sufficient resources to contribute to a fine-tuning phase of the whole model.

## 3  The Necessity of Joint Router-Expert Training

To motivate our system, we first examine whether joint training of a gating function and experts is necessary for such specialization to emerge. This question is nontrivial, as it remains unclear whether decoupled training can support effective expert allocation in the absence of explicit task boundaries. We provide a theoretical justification for this necessity by studying a simplified scenario with top-1 routing and linear experts on Gaussian input data.

Formally, consider a ground-truth orthonormal list $\mathbf{v}_1^\star, \ldots, \mathbf{v}_m^\star \in \mathbb{R}^d$. This partitions $\mathbb{R}^d$ into $m$ subsets, denoted by:

$$\mathcal{C}_j = \left\{ \mathbf{x} \in \mathbb{R}^d : \mathbf{x}^\top \mathbf{v}_j^\star \geq \max_{\ell \in [m]} \mathbf{x}^\top \mathbf{v}_\ell^\star \right\}; \quad \forall j \in [m].$$

We consider learning a function $f^\star$ that maps the input space $\mathbb{R}^d$ to labels in $\mathbb{R}$ that depends on which region $\mathcal{C}_j$ an input data $\mathbf{x}$ is sampled from. In particular, we consider there exists another orthonormal list $\mathbf{w}_1^\star, \ldots, \mathbf{w}_m^\star$ that connects the input data $\mathbf{x}$ with the output data $y \in \mathbb{R}$ by:

$$y = \sum_{j=1}^{m} \mathbb{I}\{\mathbf{x} \in \mathcal{C}_j\} \cdot \mathbf{x}^\top \mathbf{w}_j^\star; \quad \mathbf{x} \sim \mathcal{N}\left(\mathbf{0}, \tfrac{1}{\sqrt{d}} \mathbf{I}_d\right).$$

Our goal is to learn an MoE model parameterized by $\boldsymbol{\theta} = \{(\mathbf{v}_j, \mathbf{w}_j)\}_{j=1}^{m}$ where $\{\mathbf{v}_j\}_{j=1}^{m}$ are the parameters of the gating function, and $\mathbf{w}_j$ is the parameter of the $j$th expert:

$$f(\boldsymbol{\theta}, \mathbf{x}) = \sum_{j=1}^{m} \mathbb{I}\left\{ \mathbf{v}_j^\top \mathbf{x} \geq \max_{\ell \in [m]} \mathbf{v}_\ell^\top \mathbf{x} \right\} \mathbf{w}_j^\top \mathbf{x}.$$

In particular, given an input vector $\mathbf{x}$, we define the output of the $j$th expert as $\mathbf{w}_j^\top \mathbf{x}$, and the vector $\mathbf{V}\mathbf{x} = [\mathbf{v}_1^\top \mathbf{x}, \ldots, \mathbf{v}_m^\top \mathbf{x}]$ the gating output. Therefore, the indicator function $\mathbb{I}\{\mathbf{v}_j^\top \mathbf{x} \geq \max_{\ell \in [m]} \mathbf{v}_\ell^\top \mathbf{x}\}$ can be seen as the top-1 routing based on the gating function's output. We formulate the training of the MoE model as minimizing the following MSE loss:

$$\mathcal{L}(\boldsymbol{\theta}) = \frac{1}{2} \mathbb{E}_{(\mathbf{x}, y)} \left[ (f(\boldsymbol{\theta}, \mathbf{x}) - y)^2 \right] = \frac{1}{2} \mathbb{E}_{(\mathbf{x}, y)} \left[ \sum_{j=1}^{m} \mathbb{I}\left\{ \mathbf{v}_j^\top \mathbf{x} \geq \max_{\ell \in [m]} \mathbf{v}_\ell^\top \mathbf{x} \right\} (\mathbf{w}_j^\top \mathbf{x} - y)^2 \right].$$

Note there exists $\boldsymbol{\theta}^\star$ such that $\mathcal{L}(\boldsymbol{\theta}^\star) = 0$, by defining $\boldsymbol{\theta}^\star = \{(\mathbf{v}_j^\star, \mathbf{w}_j^\star)\}_{j=1}^{m}$. To this end, the theorem below characterizes the limitation of the disjoint training of experts and gating parameters.

**Theorem 1.** *Assume that $m \leq \sqrt{d}$. Assume the gating parameters are initialized according to $\mathbf{v}_{j,0} \sim \mathcal{N}(\mathbf{0}, \gamma^2 \mathbf{I}_d)$ independently for all $j \in [m]$. Let $\hat{\boldsymbol{\theta}}$ be the parameter obtained by training the experts first, and then the gating function. Formally, let $\hat{\boldsymbol{\theta}} = \{(\hat{\mathbf{w}}_j, \hat{\mathbf{v}}_j)\}_{j=1}^{m}$ be defined as:*

$$\{\hat{\mathbf{w}}_j\}_{j=1}^{m} = \arg\min_{\{\mathbf{w}_j\}_{j=1}^{m}} \tfrac{1}{2} \mathbb{E}_{(\mathbf{x}, y)} \left[ \sum_{j=1}^{m} \mathbb{I}\left\{ \mathbf{v}_{j,0}^\top \mathbf{x} \geq \max_{\ell \in [m]} \mathbf{v}_{\ell,0}^\top \mathbf{x} \right\} (\mathbf{w}_j^\top \mathbf{x} - y)^2 \right] \tag{1}$$

$$\{\hat{\mathbf{v}}_j\}_{j=1}^{m} = \arg\min_{\{\mathbf{v}_j\}_{j=1}^{m}} \tfrac{1}{2} \mathbb{E}_{(\mathbf{x}, y)} \left[ \sum_{j=1}^{m} \mathbb{I}\left\{ \mathbf{v}_j^\top \mathbf{x} \geq \max_{\ell \in [m]} \mathbf{v}_\ell^\top \mathbf{x} \right\} (\hat{\mathbf{w}}_j^\top \mathbf{x} - y)^2 \right] \tag{2}$$

*Then with probability at least $1 - \exp\left(-\Theta\left(m^2\right)\right)$, we have that $\mathbb{E}_{\{\mathbf{v}_{j,0}\}_{j=1}^m}\left[\left\|\hat{\mathbf{w}}_j - \mathbf{w}_{j'}^\star\right\|_2\right] \geq \Omega\left(1\right)$ for all $j, j' \in [m]$, and that $\mathbb{E}_{\{\mathbf{v}_{j,0}\}_{j=1}^m}\left[\mathcal{L}\left(\hat{\boldsymbol{\theta}}\right)\right] \geq \Omega\left(1\right)$.*

**Remarks:** Here, (1) describes the process of training the experts' parameters $\{\mathbf{w}_j\}_{j=1}^m$, while fixing the gating's weights at initialization $\{\mathbf{v}_{j,0}\}_{j=1}^m$. After the experts parameters are learned, (2) trains the gating's parameters, while fixing the experts' weights at $\{\hat{\mathbf{w}}_j\}_{j=1}^m$ learned in (1). What Theorem 1 demonstrates is that training experts first followed by the training of gating function with the trained expert weights frozen incurs a test loss that is lower bounded by a constant factor. Note that $\mathbf{w}_j^\star$ has unit norm; then, by the standard concentration property of Gaussian random vectors, almost all labels $y$ should have constant magnitude. Therefore, even a trivial choice of the parameters by choosing all $\mathbf{v}_j$ and $\mathbf{w}_j$ to be 0 will incur a constant loss. This means that the joint training of the expert and the gating's parameters will not improve upon the trivial choice of the parameters by more than a constant factor, motivating the need to perform joint training of the two parts. The proof of Theorem 1 –with all the details of the assumptions made– is provided in Appendix L.

Guided by this result, we now turn to the design of our proposed system, DDOME. DDOME operationalizes this principle in a decentralized learning setup and leverages data heterogeneity across different nodes to achieve expert specialization, even within a single domain.

## 4 Overview of DDOME

**System components.** Our system is depicted in Figure 1, with components grouped across the server (using purple boxes) and client (using cyan boxes).

**Server-side.** Parts **(a)**, **(b)**, **(f)**, **(h)** in Figure 1. The server maintains two key modules: $i$) a pool of $M$ experts (MoE module), each initialized with the same architecture (e.g., TinyBert); and $ii$) a gating function that ranks experts based on data characteristics. The experts can be randomly initialized or be pretrained; their weights are denoted as $\mathbf{W}_i$, for $i \in [M]$. The gating function is a small MLP with parameters $\mathbf{W}_r$ that outputs a relevance score for each expert based on input representations.

**Client-side.** Parts **(c)**, **(d)**, **(e)**, **(g)** in Figure 1. Each client has access to the same frozen, pretrained *common expert* that serves as a local feature extractor. This common expert transforms raw inputs into embeddings, which are further fed into server's gating function. *Note that the common expert is not retrained during our procedure but used as an embedding mechanism.* The result of the gating function is a sparse-enforced selection of experts. The selected experts, say experts $i$ and $j$, are dispatched to the client to be locally trained. Each client jointly updates the assigned experts' parameters –denoted as $\mathbf{W}_{i \in \mathbf{e}_s}$, with a slight abuse of notation– along with the gating function $\mathbf{W}_r$. Finally, the updated parameters $\mathbf{W}_{i \in \mathbf{e}_s}$ and $\mathbf{W}_r$ are sent back to the server to be aggregated with other updates coming from other participating clients during the training round.

**The gating function.** The gating function consists of two components: $i$) a pretrained common model that acts as a feature extractor, converting each client's local data samples into embeddings. By design, our gating function should be model agnostic to the pretrained common expert. This model is treated as a fixed, black-box encoder and does not require further training. $ii$) an *expert-ranking network*, which takes the extracted embeddings as input and outputs a relevance score for each expert. This network is updated locally by each participating client in that training round using its own data. The expert-ranking network aggregates scores across local data samples and selects the most relevant experts to be sent to each active client. This sparse-enforced, per-round selection ensures that each client only interacts with a targeted subset of experts, promoting both efficiency and specialization.

**Expert-Client relationship.** To encourage early and stable specialization, we introduce a client activation strategy, called *anchor clients*. This strategy serves two key purposes: $i$) it guides experts toward meaningful initial specialization, and $ii$) it helps the gating function better characterize each expert's behavior. Specifically, given $M$ experts in the system, we pre-select $M$ clients (out of a much larger $S \gg M$) to act as anchor clients. Each anchor client is persistently aligned with a specific expert in a one-to-one fashion and is activated more frequently than regular clients. During training, these clients are optimized with an independent loss. This design encourages consistent, distinctive expert behaviors; this strategy improves convergence stability and expert diversity (Section 5).

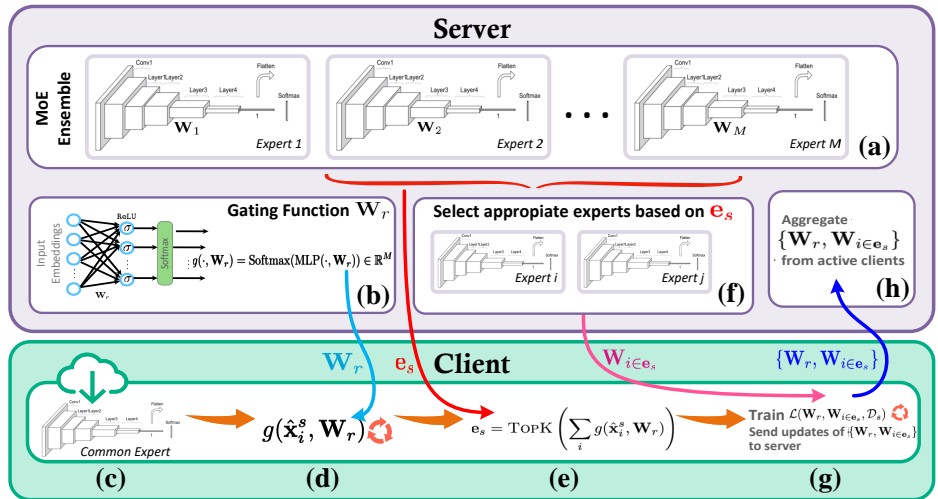

Figure 1: For each client: $i$) The server uses the gating function to select a subset of experts based on the local data distribution (parts **(a)**, **(b)**, **(d)**, **(e)**); $ii$) The client updates expert and gating function's weights (part **(g)**) and sends these back to the server. $iii$) the server aggregates and update the new weights (part **(h)**). The above are repeated for all FL rounds.

## Module details

**Pretraining.** Each client utilizes a pretrained common expert with parameters $\mathbf{W}_c$. E.g., such an expert could be a pretrained model on ImageNet for image classification purposes. We restrict our methodology such that: $i$) we ensure our algorithm is agnostic to both the common expert's architecture and performance; $ii$) we assume only access to the common expert's embedding capabilities; and $iii$) we do not modify/retrain the common expert. *The common expert is sent to/downloaded by all clients only once, before training.* For client $s$, we perform one-time inference on all local data using the common expert and store the corresponding output features, noted as $\hat{\mathbf{x}}_i^s$ for each $\mathbf{x}_i \in \mathcal{D}_s$.

**The set of expert models.** Our methodology involves $M$ experts, each being an independent model of the same architecture.[5] For the $i$-th expert, $i \in [M]$, we denote its parameters as $\mathbf{W}_i$ and the corresponding model function as $f(\cdot, \mathbf{W}_i)$. See also Figure 1**(a)**. We consider two cases in our experiments for completeness: The $M$ experts are randomly initialized in our image classification experiments to provide full plasticity during training, while for text classification the $M$ experts are initialized using weights transferred from a pretrained TinyBERT model. In each round, different subsets of experts are selected to be communicated to and updated by active clients based on their local data (see Figure 1**(f)**). Per round, the updated experts are sent back to the server to be aggregated before the next round starts; see Figure 1**(h)**.

**The gating function mechanics.** We randomly initialize an expert-ranking network with parameters $\mathbf{W}_r$. This is a small-scale, two-layer MLP that, for each active client, takes embeddings from the common expert and predicts a relevance score over all $M$ experts. Specifically, for client $s$, we denote the score as $g(\hat{\mathbf{x}}_i^s, \mathbf{W}_r) \in \mathbb{R}^M$, for the $i$-th data sample, based on:

$$g(\hat{\mathbf{x}}_i^s, \mathbf{W}_r) = \texttt{Softmax}(\texttt{MLP}(\hat{\mathbf{x}}_i^s, \mathbf{W}_r)) \in \mathbb{R}^M.$$

The final decision on the top-$K$ experts is made via the rule:

$$\mathbf{e}_s = \texttt{TopK}\Big( \sum_i g(\hat{\mathbf{x}}_i^s, \mathbf{W}_r) \Big),$$

over all embedded local data samples $\hat{\mathbf{x}}_i^s$, $i \in [|\mathcal{D}_s|]$ where the $\texttt{TopK}(\cdot)$ function selects the dominating experts, based on the current state of $\mathbf{W}_r$ and $\{\hat{\mathbf{x}}_i^s\}_{i=1}^n$.

**The "anchor clients" mechanism.** Given $M$ experts, we designate a subset of $M$ clients, with roughly distinct local data distributions, as *anchor clients*. [6] All remaining clients are referred to as *normal clients*. Each anchor client is pre-assigned to a unique expert in a one-to-one relationship,

---

[5]This choice is made for simplicity. Diverse architectures per expert are left for future work.
[6]The anchor client selection process is detailed in Appendix A.

**Algorithm 1** DDOME

**Parameters**: $T$ rounds, $S$ training clients, $U$ testing clients, $M$ experts, $\ell_1$ local iterations, experts' function and parameters $f(\cdot, \mathbf{W}_i)$, gating function's function and parameters $g(\cdot, \mathbf{W}_r)$, common expert's parameters $\mathbf{W}_c$.

---

♠ **Pretraining** ♠

Send $\mathbf{W}_c$ to all clients;
// `Data embedding`
**for** $s = 1, \ldots, S$ **do**
    $\hat{\mathbf{x}}_i^s = f(\mathbf{x}_i^s, \mathbf{W_c})$
**end for**

---

♠ **Training** ♠

**for** $t = 0, \ldots, T-1$ **do**
    Activate $N_a$ anchor and $N_c$ normal clients;
    Send $g(\cdot, \mathbf{W}_r)$ to all activated clients;
    **for** $q = 1, \ldots, N_a$ **do**
        Send expert $\mathbf{W}_{\mathbb{I}_q}$ to client $q$;
        **for** $l = 1, \ldots, \ell_1$ **do**
            $\mathbf{W}_r^q = \mathbf{W}_r^q - \eta \frac{\partial \mathcal{L}(\mathbf{W}_r, \mathcal{D}_q)}{\mathbf{W}_r^q}$;
            $\mathbf{W}_{\mathbb{I}_q} = \mathbf{W}_{\mathbb{I}_q} - \eta \frac{\partial \mathcal{L}(\mathbf{W}_{\mathbb{I}_q}, \mathcal{D}_q)}{\mathbf{W}_{\mathbb{I}_q}}$;
        **end for**
    **end for**
**end for**

---

**for** $s = 1, \ldots, N_c$ **do**
    Select a subset of experts $\mathbf{e}_s$ for client $s$;
    $\mathbf{e}_s = \text{TOPK}(\sum_{i=1}^{|\mathcal{D}_s|} g(\hat{\mathbf{x}}_i^s, \mathbf{W}_r))$;
    Send experts $\mathbf{W}_i, i \in \mathbf{e}_s$ to client $s$;
    **for** $l = 1, \ldots, \ell_1$ **do**
        $\mathbf{W}_r^s = \mathbf{W}_r^s - \eta \frac{\partial \mathcal{L}(\mathbf{W}_r, \mathbf{W}_{i \in \mathbf{e}_s}, \mathcal{D}_s)}{\mathbf{W}_r^s}$;
        $\mathbf{W}_{i \in \mathbf{e}_s} = \mathbf{W}_{i \in \mathbf{e}_s} - \eta \frac{\partial \mathcal{L}(\mathbf{W}_r, \mathbf{W}_{i \in \mathbf{e}_s}, \mathcal{D}_s)}{\mathbf{W}_{i \in \mathbf{e}_s}}$;
    **end for**
**end for**
// `Send to server for aggregation`
$\mathbf{W}_r = \text{Aggregate}(\mathbf{W}_r^q, \mathbf{W}_r^s), \forall q, s$;
$\mathbf{W}_i = \text{Aggregate}(\mathbf{W}_{i \in E_q}, \mathbf{W}_{i \in \mathbf{e}_s}), \forall q, s$;
**end for**

---

♠ **Testing** ♠

**for** $u = 1, \ldots, U$ **do**
    Send $g(\cdot, \mathbf{W}_r)$ and common expert, $\mathbf{W}_c$;
    $\mathbf{e}_u = \text{TOPK}(\sum_{i=1}^{|\mathcal{D}_u|} g(f(\mathbf{x}_i^u, \mathbf{W}_c), \mathbf{W}_r))$;
    Send experts $\mathbf{W}_j, j \in \mathbf{e}_u$ to client $u$;
    // `Perform inference`
    $j' = \max_{j \in \mathbf{e}_u} [g(f(\mathbf{x}_i^u, \mathbf{W}_c), \mathbf{W}_r)]_j$;
    $\hat{y}_i^u = f(\mathbf{x}_i^u, \mathbf{W}_{j'})$
**end for**

---

and we denote the index of the expert assigned to the $q$-th anchor client as $\mathbb{I}_q$. At the beginning of each communication round, a subset of $N$ clients are activated and are selected to participate, where $N \ll S$ and $S$ is the total number of clients in the system. We divide the $N$ active clients into two groups: $N_a$ anchor clients and $N_c$ normal clients, such that $N = N_a + N_c$. Anchor clients $N_a$ are sampled from the set of $M$ anchor clients, and normal clients $N_c$ from the remaining $S - M$ clients.[7]

The idea is that, since $M \ll (S - M)$, we more frequently sample the anchor clients. This frequent activation, coupled with fixed expert assignments, encourages each expert to specialize to the data distribution of its associated anchor client. In other words, experts are consistently trained on similar data distributions, fostering more stable and distinct specialization over time.

**The training process.** For both normal and anchor clients, the server sends the current copy of parameters of the gating function, $\mathbf{W}_r$. The gating function selects a subset of experts; the output $\mathbf{e}_s$ abstractly contains the set of chosen experts $\mathbf{W}_i$ for client $s$, where $i \in \mathbf{e}_s$. The server receives $\mathbf{e}_s$ and sends the parameters $\mathbf{W}_i$, for $i \in \mathbf{e}_s$, to the corresponding client $s$; this routine reduces the communication cost –as compared to existing methods [29]– and encourages expert specialization. We provide a theoretical analysis of the communication cost in Appendix K

Per training round, each normal client, using the standard cross-entropy loss, will locally update both $\mathbf{W}_r$ and $\mathbf{W}_i$'s. Formally, this amounts to (see also Figure 1, part **(g)**):

$$\mathcal{L}(\mathbf{W}_r, \mathbf{W}_{i \in \mathbf{e}_s}, \mathcal{D}_s) := \frac{1}{|\mathcal{D}_s|} \sum_{\{\mathbf{x}_j, y_j\} \in \mathcal{D}_s} \ell \left( \sum_{i \in \mathbf{e}_s} [g(\hat{\mathbf{x}}_j^s, \mathbf{W}_r)]_i \cdot f(\mathbf{x}_j, \mathbf{W}_i), y_j \right).$$

On the other hand, for an anchor client $q$, we only send the $\mathbb{I}_q$ expert to encourage expert specialization. Such an expert is trained regularly on the anchor's local distribution. Accordingly, we encourage the expert ranker network to recognize such rough specialization of the selected expert by using a simple independent loss. The two loss functions for anchor clients are as follows:

$$\mathcal{L}(\mathbf{W}_{\mathbb{I}_q}, \mathcal{D}_q) = \frac{1}{|\mathcal{D}_q|} \sum_{\{\mathbf{x}_i, y_i\} \in \mathcal{D}_q} \ell(f(\mathbf{x}_i, \mathbf{W}_{\mathbb{I}_q}), y_i), \quad \mathcal{L}(\mathbf{W}_r, \mathcal{D}_q) = \frac{1}{|\mathcal{D}_q|} \sum_{\{\mathbf{x}_i, y_i\} \in \mathcal{D}_q} \ell(g(\hat{\mathbf{x}}_i, \mathbf{W}_r), \mathbf{1}_{\mathbb{I}_q}),$$

where $\mathbf{1}_{\mathbb{I}_q}$ is the one-hot encoding indicating $\mathbb{I}_q$.

---

[7]The anchor-to-normal sampling ratio is discussed in Section 5.

After all clients finish the local training round, the server applies a simple aggregation step to average the updated copies of $\mathbf{W}_r$ and $\mathbf{W}_i$'s. *Adapting the MoE loss to this setting is non-trivial*. After expert selection, each client only observes and trains a small subset of experts ($K \ll M$), posing a major challenge for the gating function: if it selects "incorrect" experts, it may not only harm performance but also interfere with the specialization process by misaligning the experts. Despite this, we find that the gating function is able to learn effective expert assignments.

**Test-time generalization to unseen clients.** We are given new clients with unseen local data distributions during testing. We only send $K$ experts to each test client, and we cannot get access to local test data labels to perform fine-tuning. We first send $\mathbf{W}_r$ to the test client and select the top-$K$ experts, according to aggregated expert ranking score. Then, *for each test sample*, instead of using the weighted average of the output of all selected experts, we use the output of the expert with the highest expert ranking score to fully utilize the specialization of the expert. I.e., both experts might be utilized for different data samples instead of averaging their performance. See Algorithm 1.

## 5   Experiments

**The learning scenario we consider.** We focus our experiments on supervised image and text classification learning tasks within a Federated Learning (FL) setup [23, 21, 13], as FL offers a practical and representative environment for evaluating our system. We use the CIFAR data suite [17, 8] and EMNIST [4] for image tasks, and the Yahoo! Answers dataset [35] for text. Following common FL practice, we partition data by class to transform the full datasets into non-i.i.d. subsets. We assume the FL server-client protocol, where clients participate in training using local data; we assume there are 100 clients while we can only activate 10% clients per round. However, our system deviates from traditional FL implementations [23, 21, 13]; in those, one assumes a sole global model that is being shared with active clients, and updates to this model are being aggregated by the server per synchronization round. E.g., in image classification scenarios a large, ResNet-type network –like ResNet34, ResNet101, or ResNet200– could be used [9]. For our system, the "global" model consists of multiple independent models (experts), all sharing the same architecture. The pretrained common expert, architecturally identical to the task-specific experts, supports on-the-fly expert selection. In our experiments, we assume between 5 and 10 experts per deployment.

**Task and model description.** For the image classification task, we use ResNet-34 as the expert model architecture [9] for the federated CIFAR experiments, and a two-layer MLP for the federated EMNIST experiments. For text classification, we use TinyBERT [11]. In all cases, the gating function is implemented as a two-layer MLP followed by a softmax layer, which outputs relevance scores over all experts. For the image classification, clients are trained using the SGD optimizer with momentum (SGDM), with a learning rate of 0.01, momentum of 0.9, batch size of 256, and one local epoch per round. For text classification, all client models are trained using SGDM with identical hyperparameters. All clients use a batch size of 16, and one local epoch per round. The gating function is trained using an SGD optimizer; for image classification, this uses a fixed learning rate of 0.001, while for text classification the same initial learning rate is used but it incorporates a step decay learning rate schedule. The model aggregation on the server side is performed with FedAvg [24].

**System.** Experiments were conducted on different hardware setups. For image classification tasks, we used an NVIDIA RTX A6000 GPU with 46GB of VRAM. Training the default configuration with 10 experts required approximately 6 hours. For text classification tasks, we used an NVIDIA A40 GPU with 48GB of VRAM, and training the default configuration with 5 experts required over 100 hours (using 2 workers) due to the increased computational cost. Training was performed in a distributed fashion using between 2 and 10 workers.

**Dataset.** We conduct experiments on CIFAR10, CIFAR100 [17], EMNIST [4], and Yahoo! Answers [35]. For EMNIST, we use the "ByClass" split, which contains 814,255 character images across 62 unbalanced classes. For Yahoo! Answers, we randomly sample 50,000 training and 10,000 test examples to create a dataset of the same size as CIFAR, with an equal number of samples per class.[8] To increase task difficulty, we exclude the answer text and train models solely on the question title and content. For all cases, the training dataset is randomly partitioned across 100 clients. We followed the same procedure for the *anchor* clients but avoided replacement, aiming to preserve the label diversity in each subset. We establish a one-to-one mapping between these clients and the experts, corresponding to each group of labels. This path allows to $i$) have one expert available for

---

[8]Full dataset training is computationally expensive, given the hardware configuration we had.

each group; and *ii*) retain the flexibility to activate the *anchor* clients during the training rounds. The complete client distribution for all datasets is detailed in Appendix A.

**Zero-Shot Personalization.** Let us first describe the baselines to compare against:[9]

- `FedMix` [29] trains an ensemble of models adapted to the data space's sub-regions. By definition, `FedMix` sends all the experts to each client to specialize them, heavily increasing communication costs. For this implementation, we initialized the common expert from the initial pretrained model checkpoint, and we used it to embed local data in the gate function and help the routing.
- `FedAvg` [24] is the de facto approach for FL and allows a fair comparison regarding fixed communication cost. Here, we initialize the global model with the initial common expert checkpoint and aggregate the updates from all sampled clients per iteration.
- `FedProx` [22] tackles heterogeneity by introducing a regularization term that limits the distance between local/global models at the cost of additional computation overhead per round. We follow the same strategy for initialization with `FedAvg`.
- `Scaffold` [14] handles non-iidness by applying control variates for the server and clients at the expense of doubling communication cost per round compared to `FedAvg`. This method tends to become unstable during training, as previous studies have shown [20]. We follow the same strategy for initialization with `FedAvg`.
- The `Average Ensembles` [16] train two models (initialized from the common expert) as in `FedAvg`, but with different random seeds. It then combines them by averaging output probabilities. While it provides flexibility with respect to resources, it has higher inference costs.

| | | CIFAR 10 | | | | | CIFAR 100 | | | |
|---|---|---|---|---|---|---|---|---|---|---|
| Method | # Clients | Rounds | $M$ | $K$ | Acc. | Acc. | Rounds | $M$ | $K$ | Acc. | Acc. |
| Common Expert | – | – | – | – | 73% | 93% | – | – | – | 67% | 73% |
| FedMix [29] | 100 | 1250 | 2 | 2 | 31.3% | 42.9% | 2000 | 2 | 2 | 49.7% | 48.3% |
| FedAvg [24] | 100 | 1250 | – | – | 31.2% | 58.4% | 2000 | – | – | 72.9% | 74.0% |
| FedProx [22] | 100 | 1250 | – | – | 72.7% | 71.4% | 2000 | – | – | 72.8% | 74.0% |
| Avg Ensembles [16] | 100 | – | – | – | 23.9% | 53.7% | – | – | – | 72.8% | 74.1% |
| DDOME | 100 | 1250 | 5 | 2 | **91.8%** | **95.7%** | 2000 | 10 | 2 | **75.7%** | **78.6%** |

Table 1: Average zero-shot personalization score for unseen test clients on CIFAR10/100. See subfigures (d) and (b) of Figure 5 in Appendix C for statistical significance of the respective results. We use two different pretrained *common experts* as feature extractor for each dataset: a) The lower bound model at which the gating function can outperform the initial common expert accuracy, illustrated in Figure 8 in Appendix; b) The average model that represents a good accuracy that is relatively easy to achieve using ResNet-34 architecture. Sampling is performed under the scheme of $N_a = 5$ anchor and $N_c = 5$ normal clients per training round; here, $N = N_a + N_c = 10$. See Appendix E for `DDOME` gating function's effectiveness on individual samples.

Tables 1-3 summarize our findings on this setup. Whereas `FedMix` requires all experts to be transmitted to each client, i.e., $M = K$, `DDOME` allows the selection of $K$ experts, here $K = 2$, without the need to send them all. This reduces communication costs and ensures the client receives the most pertinent information from the relevant experts.

In terms of baselines, we observe that for the CIFAR datasets both behave differently. We attribute this gap to the number of classes each client holds. In the CIFAR10 scenario, each client has fewer classes, which can amplify the model drift problem in all baselines. Furthermore, `FedAvg`'s performance deteriorates sharply when we test it on the new CIFAR10 clients that were not used for training due to the heterogeneous data distribution during training and then in the testing phase. Similarly, `Average Ensembles` faces a performance ceiling, as the ensembles inherit the limitations of the `FedAvg` aggregation method. On the other hand, `FedProx` can surpass the initial performance of the common expert for the CIFAR100 scenario but degrades quickly when using few labels per client, as in the CIFAR10 setup. To the best of our ability, we attempted multiple hyperparameter settings for `Scaffold`, yet we were unable to produce a useful model under this distribution; it became unstable during training (achieving only 10% for CIFAR10 / <5% for CIFAR100). Further comparison against domain adaptation methods, as in `FedADG` [34] and `FedSR` [26], is shown in Appendix I; for the cases we consider, we observe that current implementations are bound to having a small number of clients to perform competitively. These trends are not limited to vision tasks.

---

[9]Note that we are aware that there are dozens more generic FL algorithms to compare against; yet, our aim is to provide a proof-of-concept for our methodology on training and selecting the right experts in such a setting.

On the Yahoo! Answers dataset (Table 2), we observe a similar pattern. All baselines struggle to consistently outperform the pretrained common expert, with `FedAvg` again showing limited generalization. However, unlike CIFAR10, the severity of model drift is reduced—likely due to clients having access to more diverse text samples per class. Despite this, the limitations `FedAvg` and `Average Ensembles` persist. As in CIFAR100, `FedProx` slightly outperforms the common expert in one configuration, but fails to do so consistently.

| | | | Yahoo! Answers | | | |
| Method | # Clients | Rounds | $M$ | $K$ | Acc. | Acc. |
|---|---|---|---|---|---|---|
| Common Expert | – | – | – | – | 61% | 69% |
| FedMix | 100 | 2000 | 2 | 2 | 58.08% | 59.46% |
| FedAvg | 100 | 2000 | – | – | 60.75% | 60.18% |
| FedProx | 100 | 2000 | – | – | 60.15% | 61.18% |
| Avg Ensembles | 100 | – | – | – | 62.42% | 69.36% |
| DDOME | 100 | 2000 | 5 | 2 | **64.55%** | **74.90%** |

Table 2: Average zero-shot personalization score for unseen test clients on Yahoo! Answers. The structure of the table follows that of Table 1.

For data diversity, we report results on the EMNIST dataset; please refer to Table 3 for more information. We note that we have also considered lower than 73% accuracy for the common expert (e.g., 67%). Yet, such an initial performance was too low to improve further using any of the methods in comparison. This led to the inclusion of the 73% and 80% cases. This highlights the importance of the common expert in our framework, underlying that our methodology does not "magically" work for all cases. Still, proper preparation is needed to obtain favorable performance.

The global accuracy reported at the end of training demonstrates the effectiveness and consistency of `DDOME` in all datasets, with significantly better performance than other algorithms. Please refer to Appendix B for a detailed end-to-end performance of the methods in Table 1 under different clients' distribution.

| | | | EMNIST | | | |
| Method | # Clients | Rounds | $M$ | $K$ | Acc. | Acc. |
|---|---|---|---|---|---|---|
| Common Expert | – | – | – | – | 73% | 80% |
| FedMix | 100 | 2000 | 2 | 2 | 9.4% | 15.9% |
| FedAvg | 100 | 2000 | – | – | 72.1% | 72.1% |
| FedProx | 100 | 2000 | – | – | 72.0 % | 72.0% |
| Avg Ensembles | 100 | – | – | – | 74.5% | 74.4% |
| DDOME | 100 | 2000 | 10 | 2 | **79.9%** | **80.5%** |

Table 3: Average zero-shot personalization score for unseen test clients on EMNIST. The structure of the table follows that of Table 1. See Figure 6 in Appendix D for statistical significance of the respective results.

**Additional ablation studies and experiments.** Appendix F contains thorough ablation studies on the initial conditions of the common expert and how it boosts the performance, and the value of anchor clients and their ratio with normal clients. Appendix E contains assessment of the `DDOME` gating function's effectiveness on individual samples. Appendix G considers the incremental learning scenario, where either the pool of clients dynamically increases over time or changes over time. Appendix H considers the case where $M = K$ and compares `FedMix` versus `DDOME`. Appendix I compares `DDOME` against domain generalization methods. Finally, Appendix J discusses how `DDOME` differ from other clustering-based methods applied on similar scenarios.

# 6 Broader impacts and limitations

**Broader impacts.** Our work advances efficient and adaptive specialization of Mixture-of-Experts models, enabling personalized ML in decentralized environments. Positive impacts include privacy-preserving FL and resource-efficient personalized applications, reducing computational and communication costs. However, we acknowledge potential risks, such as exacerbating fairness issues if expert specialization amplifies biases inherent in heterogeneous data. Future deployment should carefully monitor and mitigate such risks.

**Limitations.** Despite the practical and theoretical promise, our study exhibits several limitations that could inform future research directions:

- *Dependence on pretrained common experts:* The success of our approach relies heavily on the availability of an effective pretrained common expert (see Appendix F).

- *Communication overhead:* While `DDOME` reduces communication costs relative to sending all experts to clients, it still incurs higher communication overhead compared to traditional single-model methods.

- *Expert initialization and diversity:* Our experiments indicate that expert diversity at initialization significantly impacts specialization effectiveness. We primarily evaluated homogeneous expert architectures; further study is necessary to understand the impacts of architectural heterogeneity.
- *Limited scalability testing:* Current experimental setup tests up to a moderate number of clients.
- *Complexity of gating mechanism tuning:* Finding optimal gating configurations might become computationally expensive as scale and diversity increase.

## 7   Conclusions

In this work, we investigated how joint training of gating mechanisms and experts can enable adaptive specialization in Mixture-of-Experts (MoE) frameworks under single-domain, heterogeneous data settings. We introduced DDOME (*Dynamically Decentralized Orchestration of MoEs*), a distributed MoE architecture specifically tailored for decentralized training scenarios.

Our empirical evaluations across various datasets demonstrate that DDOME achieves state-of-the-art performance, surpassing existing FL methods by leveraging implicit data heterogeneity. Complementing empirical findings, we provided a rigorous theoretical justification for the necessity of joint training between gating and experts. Our analysis highlights that disjoint or sequential training of these components significantly limits achievable specialization, reinforcing the importance of coordinated parameter updates. Future work will explore architectural diversity among experts, scalability enhancements, and extensions to multi-domain settings.

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

# A   Clients distribution

We created a federated version of the datasets by introducing two partitioning strategies to split the samples across 100 clients:

- *Quantity-based label imbalance*: Each client holds data samples of $K$ labels. We first randomly assign $K$ different labels to each client. Then, per label, we randomly assign samples to clients along with labels (with replacement). This way, the number of different labels for each client is fixed. For the CIFAR100 dataset, we use $K = 10$. For the CIFAR10 and Yahoo! datasets, we use $K = 4$.

    **Anchor clients:** We followed the same method to create the anchor clients, except we prevented replacement when randomly selecting the labels. This way, we created a) 5 anchor clients with $K = 2$ on CIFAR10 and Yahoo! and b) 10 anchor clients with $K = 10$ on the CIFAR100 dataset.

- *Distribution-based on label imbalance*: We simulated the label imbalance of each client by allocating a portion of the samples (with replacement) of each label according to the Dirichlet distribution ($\alpha = 0.1$). As illustrated in Figure 2, the test clients are randomly unseen combinations of $K$ labels that never appear during training.

    **Anchor clients:** We use the same Dirichlet distribution ($\alpha = 0.1$) to randomly create a) 5 anchor clients on CIFAR10 and b) 10 anchor clients on the CIFAR100 dataset.

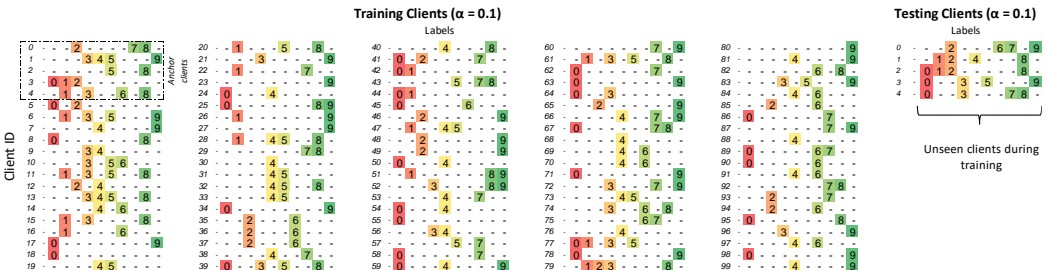

Figure 2: Example of distribution-based label imbalance partition on CIFAR10 dataset ($\alpha = 0.1$)

We conducted a simulation of a federated version of 100 clients on the EMNIST dataset, using the "ByClass" split. This split presents a greater challenge than the CIFAR10/100 datasets, as some classes have a much larger number of samples, resulting in 62 unbalanced classes. The clients were created in accordance with the *Quantity-based label imbalance* approach, with $K = 5$ .

Note that for test users, we do not repeat any distribution from the training clients; this way, we create an example where the distribution of the images over all users is different.

# B   `DDOME` **end-to-end performance**

## B.1   Quantity based strategy

We begin to evaluate the performance of our method and baselines by measuring the zero-shot personalized model accuracy on several unseen test clients with a Quantity-based label imbalance distribution strategy, as explained in Appendix A. The results are illustrated in Figure 3.

In Figure 3, we can observe that `FedAvg` cannot keep improving once it's initialized from the pretrained checkpoint. This surprising result stems from three major issues: $i)$ the clients' learning rate parameters are inconsistent with previous training, $ii)$ the heterogeneous data distribution on the training clients introduces a high degree of model variability, and $iii)$ the pretrained expert struggles to improve or adapt to the federated distribution. Moreover, implementing `FedProx` required careful fine-tuning of the $\mu$ parameter to achieve good accuracy and fast convergence. On the other hand, despite trying multiple hyperparameter settings, we could not produce a useful model using the `Scaffold` method; it became unstable during training and often collapsed or got stuck in a poor model. This suggests that our method is more robust than these baselines in the current setup.

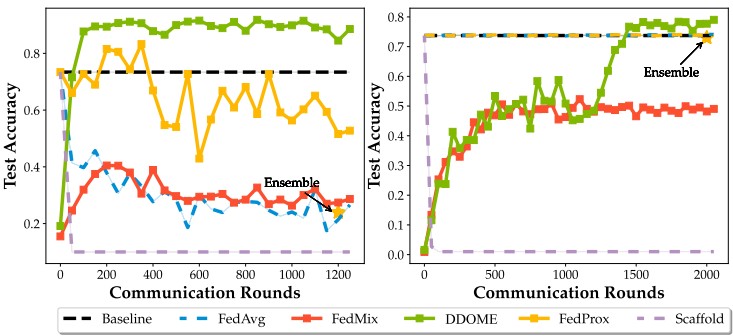

Figure 3: `DDOME` on CIFAR10 (left) and CIFAR100 (right) datasets, against `FedMix`, `FedAvg` and `Average Ensembles` based on Table 1, using an initial common expert of 73% accuracy.

## B.2  Distribution based strategy

Using the distribution-based strategy –detailed in Appendix A– we implement two additional challenging scenarios, where further heterogeneity and complexity are inserted via labels distribution: $i$) we use the Dirichlet probability rule to generate skewed and imbalanced label distributions, mimicking real-world applications; $ii$) we relax the assumption of disjoint labels for the *anchor* clients and allow label overlap, creating a more complex scenario, given that experts are initialized from scratch.

|  | CIFAR 10 | CIFAR 100 |
|---|---|---|
| Common Expert | 73.39% | 73.73% |
| FedAvg | 51.3% | 73.6% |
| FedProx | 52.8% | 73.6% |
| Scaffold | 10.0% | 01.0% |
| FedMix | 29.8% | 65.3% |
| DDOME | **80.8%** | **77.8%** |

Table 4: Best global test accuracies from the last ten evaluation rounds reported on different non-iid algorithms under Dirichlet distribution ($\alpha = 0.1$).

Table 4 indicates `DDOME` leverages the common expert's original 73% accuracy to reach up to 80% accuracy, even on highly skewed scenarios. While heterogeneity should decrease the overall performance, `DDOME` outperforms the methods under comparison, where experts learn to better generalize to unseen data.

Further, in Figure 4, we show the test accuracy envelope curves for all the algorithms under consideration. It is clear that `DDOME` show superior performance throughout execution, often surpassing the text accuracy of the common expert, which is already sufficiently trained, and there is limited space for improvement. This figure also shows the behavior of the models during training: even `DDOME` shows the variability of test accuracy over iterations, indicating that keeping the model at the very end of the execution might not always be the best practice.

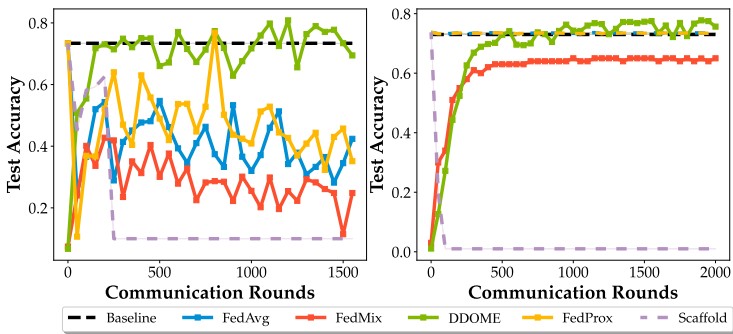

Figure 4: Evaluation of different non-iid algorithms under Dirichlet distribution ($\alpha = 0.1$) on CIFAR10 dataset.

## C  Performance under different sampling ratios

There is an initial degree of randomness in the gating function. During the first couple of iterations, it sends random top-$K$ experts to each client while the experts learn to specialize in the different regions of the label space. However, we found a way to keep consistency during these initial rounds: through the *anchor clients*. Figure 9 shows that by introducing at least 30% *anchor* clients during each round, we can ensure a balance against the wrong selection of the gating function by letting them act as *regularizers*. Additionally, Figure 5 shows the impact on the performance when we remove the anchor clients rule from sampling and allow only random selection from the pool of available clients. It is clear from Figure 5 that a "warming-up" phase is necessary for DDOME: using a sufficient number of anchor clients, one achieves stability and better final accuracy, by warm-starting the system using more specialized clients.

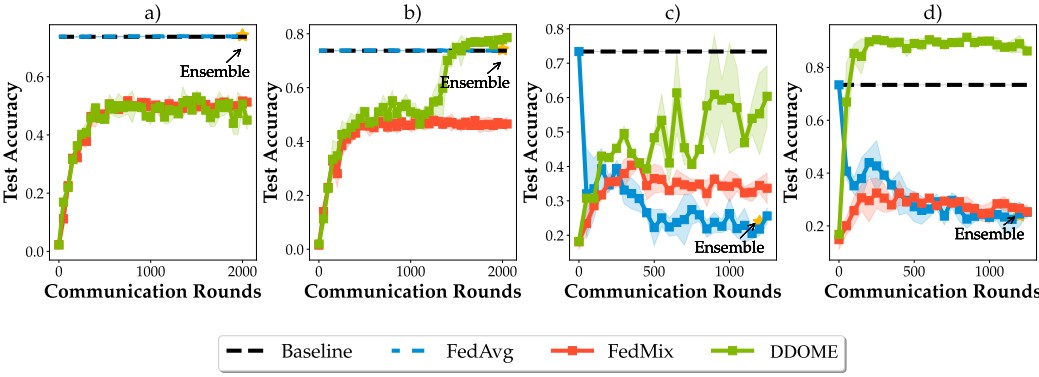

Figure 5: Global testing accuracy for CIFAR100 (a-b) and CIFAR10 (c-d) datasets on two different sampling strategies: a) + c): 10 random clients without replacement per iteration. b) + d): 5 random *anchor* clients + 5 normal clients without replacement per iteration along different methods.

## D  EMNIST Byclass statistical significance results

We evaluate a two-layer MLP on the federated EMNIST dataset, running each algorithm for 1000 communication rounds across three different initialization seeds. Results show that DDOMe consistently surpasses the performance of the initial common expert, demonstrating its robustness and effectiveness across varying accuracy levels.

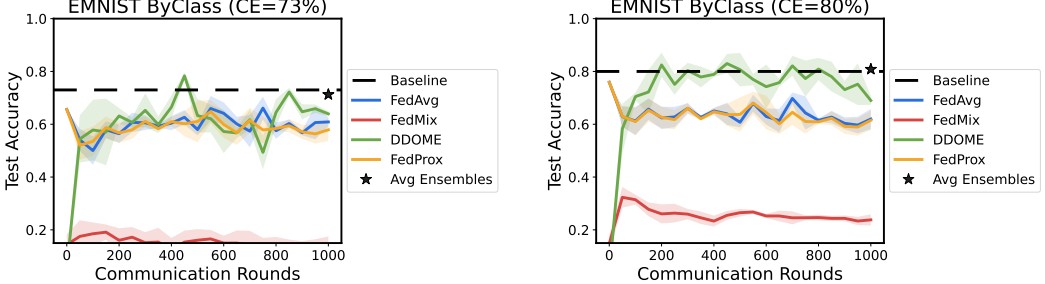

Figure 6: Statistical significance results for EMNIST dataset using a common expert with 73% accuracy (left subfigure) and 80% accuracy (right subfigure).

## E  Gating function Per-Sample Performance

After training, we thoroughly evaluate our gating function, using the checkpoints trained with the 73% *common expert* on CIFAR10 and CIFAR100 datasets on the DDOME algorithm. Our fine-grained evaluation demonstrates that our gating function can analyze the characteristics of each unseen test client's local sample and adaptively select a subset of experts that match those characteristics. This is crucial in ensuring that our gating function can generalize well to new data. After selecting the top-$K$ experts, the gating function chooses the highest score/confidence expert to predict each test

data sample. Our results, reported in Table 5, show that our gating function can achieve high accuracy on the selection.

| CIFAR100 | | | | CIFAR10 | | | |
|---|---|---|---|---|---|---|---|
| Client | Incorrect | Correct | Error Rate | Client | Incorrect | Correct | Error Rate |
| 0 | 278 | 722 | 27.8% | 0 | 227 | 3773 | 5.7% |
| 1 | 281 | 719 | 28.1% | 1 | 122 | 3878 | 3.1% |
| 2 | 263 | 737 | 26.3% | 2 | 563 | 3437 | 14.1% |
| 3 | 251 | 749 | 25.1% | 3 | 103 | 3897 | 2.6% |
| 4 | 261 | 739 | 26.1% | 4 | 78 | 3922 | 2.0% |
| 5 | 309 | 691 | 30.9% | | | | |
| 6 | 260 | 740 | 26.0% | | | | |
| 7 | 285 | 715 | 28.5% | | | | |
| 8 | 255 | 745 | 25.5% | | | | |
| 9 | 267 | 733 | 26.7% | | | | |
| Average Error Rate | | | 27.1% | Average Error Rate | | | 5.5% |

Table 5: Evaluation per-sample level on CIFAR10 and CIFAR100 datasets.

## F   Ablation studies

**Ablation study: Initial common expert impact.** We study performance tradeoffs when utilizing different common experts for the gating function decisions. Our findings indicate that the amount of training allocated in the initial *common expert* has a critical effect on the overall performance of DDOME. For example, suppose the gating function uses a poor common expert for training. In that case, it can lead to poor performance (collapses to selecting a single expert) and, therefore, unable to improve beyond the baseline.

Figures 7-8 show that the breakpoint of the gating function for the CIFAR100 dataset is approximately 66% accuracy by the *common expert*. In Figure 8, it becomes clear that a significant cause of this breakpoint is that the experts cannot surpass the common expert's initial accuracy. This is attributed to the lack of an adequate selection of experts, which is essential for the gradient updates of each expert to be aligned with the same part of the task. Figure 7 also reveals the following: the 67% case, given a few more iterations, can match the performance of the 73% case. This suggests a "phase-transition" might exist, where more effort (i.e., communication) is needed to improve beyond the common expert's performance. This implies that the performance of DDOME depends on the quality of the experts.

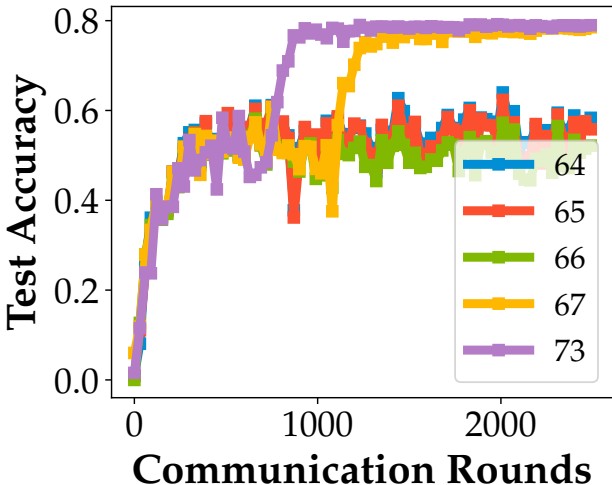

Figure 7: DDOME's performance on CIFAR100 dataset, using different initial accuracy for *common expert* (legends of the plot); the setup in Table 1 is used.

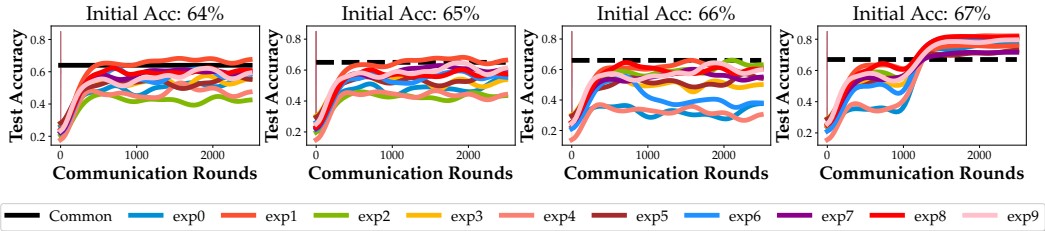

Figure 8: Zero-shot personalization accuracy per expert during training on CIFAR100.

**Ablation study: Common expert boosts experts' performance.** To test this, we initialized each expert from the common expert and continued training for 2000 rounds. In Table 6, we observe the final score of each method. Surprisingly, for DDOME, it takes a few more rounds to overcome the baseline than when the experts are initialized from scratch. This is because the pretrained model is optimal for the entire dataset. To successfully specialize experts, retraining the model on the specific subset of labels is necessary.

| Common Expert | 73.73% |
| --- | --- |
| FedMix | 73.78% |
| FedAvg | 73.99% |
| Average Ensembles | 74.10% |
| DDOME | **83.27%** |

Table 6: Average zero-shot accuracy for CIFAR100 after 2000 rounds.

We also plot in Figure 8 the performance of each expert (denoted as expX) over the communication rounds for different initial accuracies of the common expert. It is evident that, for our setting, using a common expert with an accuracy below 67% We can outperform the other methods once the gating function can utilize a slightly better common expert.

**Ablation study: The "anchor/normal" client ratio.** The sampling scheme must be carefully studied to ensure the best performance of DDOME. Each expert has a distinct distribution; i.e., their local objectives only align with a particular subset of labels. Ensuring consistency in the experts' updates is essential to prevent them from drifting away from their own "task". As mentioned, we assume we have some control over the activation of the clients during training.

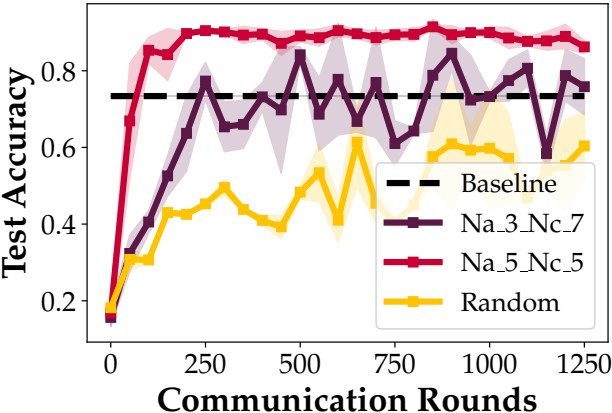

Figure 9: DDOME_Na_X_Nc_Y means that $\frac{X}{Y} = \frac{N_a}{N_c}$, and $N = N_a + N_c$. 30% anchor/normal client ratio is enough to match baseline accuracy. However, the model becomes more inconsistent by converging more slowly.

Our solution is the use of *anchor* clients, whose primary purpose is to act as regularizers, ensuring consistency in the expert updates during training. To find the optimal ratio of anchor/normal clients $\frac{N_a}{N_c}$, we conduct experiments varying this ratio; see Figure 9. Sampling half of the clients per round as *anchor* quickly surpasses the baseline of the common expert and maintains high consistency in subsequent iterations. Using a lower ratio of 30% anchor clients per round also achieved similar performance, allowing some flexibility in the sampling. Contrarily, when we sampled clients randomly from the available pool (i.e., no "anchor clients"), DDOME shows difficulty improving performance,

as experts' updates become inconsistent. Appendix C shows the end-to-end performance difference across different methods using these sampling ratios for both datasets.

# G    Incremental Learning

Incremental learning is a paradigm that aims to update and refine existing knowledge from new data rather than discarding or retraining from scratch. This can benefit scenarios where data is dynamic, scarce, or costly to acquire and where learning models must adapt to changing environments or tasks. We performed a comprehensive comparison using the same benchmarking methods in Table 1 to contrast each algorithm's learning process.

## G.1    Dynamically increase the client's pool

For this setup, we split the CIFAR100 dataset into five groups with non-overlapping labels. Each group held 20 different clients with random samples within the label range. Then, we allowed only one group of labels to be trained for 200 iterations. Afterward, we increased the pool of clients with a new group each 200 iterations, monitoring the global accuracy of the models over time. In Figure 10, we can observe that DDOME is not affected if the entire set of clients is not present from the outset; its gating function develops adaptively, without compromising its ability to capture the old distributions. In contrast, Fed-Mix drops its performance by approximately 4% compared to the original results in Table 1.

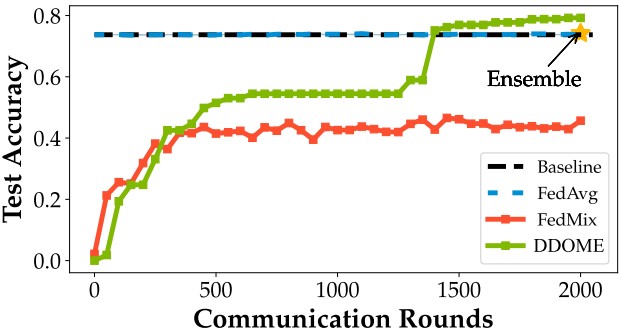

Figure 10: Incremental Learning scenario on CIFAR100, dynamically increasing the total pool of clients.

## G.2    Dynamically switch the client's pool

We employ a cyclical learning approach based on the first setup for the second scenario. Instead of simply increasing the total pool of clients, we only allow one of the five groups of clients to contribute to the training process at a time. This means that every 600 iterations, we switch the pool of available clients, allowing us to see new labels and ensuring that the labels seen during the initial iterations will never be seen again during the training process. This cyclical approach allows us to benefit from the data's diversity while ensuring that the model is constantly being exposed to new information.

Figure 11 illustrates that even when DDOME is approximately 2% below FedAvg at the end of the training, the former continues to improve. At the same time, the other methods begin to decline over the iterations. This is likely due to the anchor clients acting as regularizers to adjust the gradient directions during optimization, as the client pool presents a more complex setup. The anchor clients can provide a more stable optimization process.

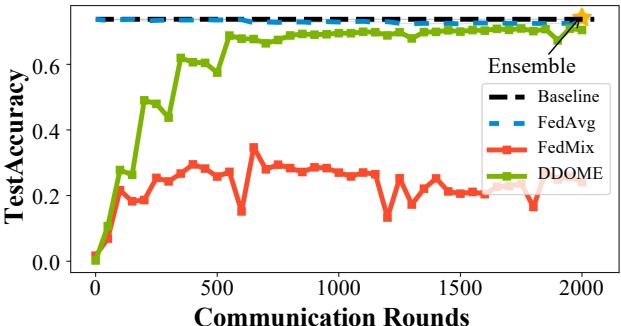

Figure 11: Incremental learning scenario on CIFAR100, dynamically switching the total pool of clients

## H Performance under matching number of experts $M = K$

We present additional experiments to compare `FedMix` vs. `DDOME`, using the same number of total experts. I.e., $M = K$ to disentangle our method's behavior under different experts. The results are shown in Table 7, where it is evident that even if we send the complete set of experts per worker, our approach performs better. Yet, the resulting model might be less accurate than the common expert, implying that malicious "interference" exists.

|  | $M = K = 2$ | $M = K = 5$ |
|---|---|---|
| Common Expert | 73.39% | 73.39% |
| FedMix | 42.76% | 43.86% |
| DDOME | **60.16%** | **75.77%** |

Table 7: Best global test accuracy reported during training on the CIFAR10 dataset under Dirichlet distribution ($\alpha = 0.1$) with a fixed number of models communicated to each client. Both methods were initialized from the same *common expert* with an initial accuracy of 73.39%.

## I Comparison against Domain Generalization Methods

Our scenario can also be framed as a *Domain Generalization* problem. Thus, we evaluate `DDOME` against state-of-art methods, such as `FedSR` [26] and `FedADG` [34], that handle robustness to distribution shifts on test-time. Results in Table 8 demonstrate that the ability of `FedADG` and `FedSR` to evaluate unseen domains is tightly bound to a small number of clients. Once we increase the underlying distribution (e.g., 100 different clients), these methods cannot exploit the cross-relationship among domains [1].

| Common Expert | 93.05% |
|---|---|
| FedSR[26] | 28.24% |
| FedADG[34] | 41.83% |
| DDOME | **87.86%** |

Table 8: Best global test accuracy reported during training on the CIFAR10 dataset using quantity-based label imbalance. We sample 10 (of 100 available) random clients during 900 iterations with replacement. All methods were initialized from the same *common expert* reported in the Table.

## J Clustering analysis

To provide a more extensive comparison of our expert models, it is essential to highlight that the core idea is not to summarize clients into several models, as many clustering-related works do. Clustering methods are limited to scenarios where clients are inherently grouped; all clients in the same group will have similar local data distributions, while clients across groups will share few data. Instead, we target a more realistic scenario, where each client has a non-iid and mixed-data distribution, making client clustering based on local distributions less meaningful. To illustrate this, we have performed an example of client clustering using K-means on local class distributions as shown in Figure 12,

where each dot represents one client and the annotated numbers are this client's two main data classes. The color represents the K-means clustering result. Clearly, clustering does not create meaningful groups of clients, and training individual experts in each group does not provide any specialization of experts.

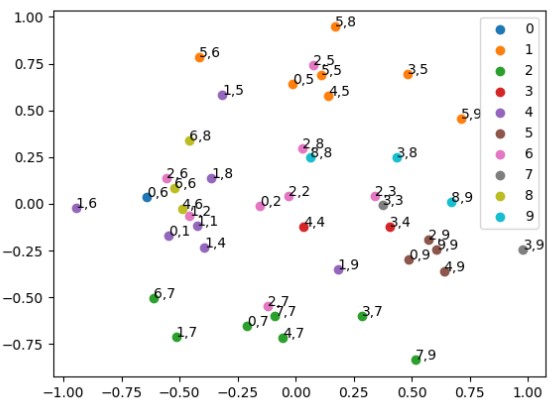

Figure 12: Clients clustering with label frequency.

# K   Theoretical communication cost analysis of DDOME and FedMix

**Variable definitions**

$M$: number of expert models.

$R$: number of communication rounds.

$S$: number of clients.

$N$: active clients.

$k$: topk experts ($k \leq M$).

$P_r$: parameters of router.

$P_e$: parameters of a single expert.

$P_c$: parameters of common expert.

$\epsilon$: size to communicate an expert index.

## K.1   Analysis of DDOME

**Communication Cost Derivation**

Total cost = initial setup + cumulative round cost.

**Initial Cost** ($C_{init}$): This one-time, server-to-client cost involves sending the common expert model to all clients $S$.

$$C_{\text{init}} = S \cdot P_c$$

**Per-Round Cost** ($C_{round}$): For each of the $R$ rounds, the cost is the sum of downlink (Server-to-Client) and uplink communication (Client-to-Server).

**Downlink Cost** ($C_{down}$): Router ($P_r$) and $k$ experts to $N$ active clients.

$$C_{down} = N(P_r + k \cdot P_e)$$

**Uplink Cost** ($C_{up}$): top-$k$ indices + updates for router + $k$ experts to $N$ active clients.

$$C_{up} = N(k \cdot \epsilon + P_r + k \cdot P_e)$$

Note: The communication here does not represent computational order.

The total cost for a single round is therefore:

$$C_{\text{round}} = C_{\text{down}} + C_{\text{up}} = N(2P_r + 2kP_e + k\epsilon)$$

**Total Communication Cost ($C_{total}$):** The total cost over $R$ rounds is the sum of the initial cost and the cumulative round costs.

$$C_{total} = C_{init} + R \cdot C_{round} = (S \cdot P_c) + R \cdot N(2P_r + 2kP_e + k\epsilon)$$

Since the parameter sizes are much larger than index sizes ($P_r, P_e \gg \epsilon$), we can approximate the total cost as:

$$C_{\text{total}} \approx (S \cdot P_c) + 2R \cdot N(P_r + kP_e)$$

## K.2 Analysis of FedMix

The FedMix algorithm communicates all experts in every round.

**Communication Cost Derivation**

**Per-Round Cost ($C_{\text{round, FedMix}}$)** In each round, every active client downloads and uploads all the models.

**Downlink Cost ($C_{\text{down, FedMix}}$):** The server sends all $M$ experts ($M \cdot P_e$) to each of the $N$ active clients.

$$C_{\text{down, FedMix}} = N(M \cdot P_e)$$

**Uplink Cost ($C_{\text{up, FedMix}}$):** Each of the $N_{\text{active}}$ clients sends back the updated parameters of the $M$ experts.

$$C_{\text{up, FedMix}} = N(M \cdot P_e)$$

**Total Communication Cost ($C_{\text{total, FedMix}}$)** The total cost is the per-round cost multiplied by the number of rounds, $R$.

$$C_{\text{total, FedMix}} = 2R \cdot N(M \cdot P_e)$$

## K.3 Limit Analysis

This analysis examines the asymptotic behavior of the communication costs under the following assumptions:

- The number of active clients is a fixed fraction of the total: $N = S/c$ for some constant $c > 0$.
- The parameter size of the initial common expert is equal to that of a single expert: $P_c = P_e$. Note: This is true in our experiments.

### K.3.1 Analysis as the number of experts $M \to \infty$

- **DDOME**:

$$\lim_{M \to \infty} C_{\text{total}} = \lim_{M \to \infty} [(S \cdot P_c) + 2R \cdot N(P_r + kP_e)] = (S \cdot P_c) + 2R \cdot N(P_r + kP_e)$$

The cost is independent of $M$, implying that the communication cost is bounded and does not increase as the number of experts grow.

- **FedMix**:

$$\lim_{M \to \infty} C_{\text{total, FedMix}} = \lim_{M \to \infty} [2R \cdot N(M \cdot P_e)] = \infty$$

The cost is a linear function of $M$. As the number of experts grows, the cost grows without bound. This means the communication cost is unbounded, making the algorithm very unscalable compared to DDOME.

### K.3.2 Analysis as Number of Rounds $R \to \infty$

We analyze the cost as the training process becomes infinitely long.

- **DDOME** The cost is of course unbounded but grows linearly with $R$. The rate of growth is given by the partial derivative with respect to $R$:

$$\frac{\partial C_{\text{total}}}{\partial R} = \frac{2S}{c}(P_r + kP_e)$$

- **FedMix** The rate of growth with respect to $R$ is:

$$\frac{\partial C_{\text{total, FedMix}}}{\partial R} = \frac{2S}{c}(MP_e)$$

**Comparison** Both costs are unbounded as $R \to \infty$. However, comparing their growth rates reveals that for $k \ll M$, DDOME grows significantly slower, making it more efficient for longer training durations.

$\frac{\frac{2S}{c}(P_r + kP_e)}{\frac{2S}{c}(MP_e)} < 1$

### K.3.3 Analysis as Number of Clients $S \to \infty$

In this case we will add the $\epsilon$ term back in, as it directly depends on $S$.

- **DDOME** The rate of growth with respect to $S$ is:

$$\frac{\partial C_{\text{total}}}{\partial S} = P_e + \frac{R}{c}(2P_r + 2kP_e + k\epsilon)$$

- **FedMix Algorithm** The rate of growth with respect to $S$ is:

$$\frac{\partial C_{\text{total, FedMix}}}{\partial S} = \frac{2R}{c}(MP_e)$$

**Comparison**. The goal is to prove that FedMix rate of change is greater. We will show that this is indeed the case under certain (realistic) conditions.

We seek to demonstrate that: $2\frac{R}{c}MP_e > P_e + \frac{R}{c}(2P_r + 2kP_e + k\epsilon)$

We proceed by rearranging this inequality to find the condition under which it holds true.

$\frac{R}{c}\left[2(M-k) - \frac{2P_r}{P_e} - \frac{k\epsilon}{P_e}\right] > 1$

$2(M-k) - \frac{2P_r}{P_e} - \frac{k\epsilon}{P_e}$, is a positive number slightly less than $2(M-k)$ and almost certainly greater than 1.

Thus, the inequality FedMix > DDOME holds true if:

$\mathbf{R} > \frac{\mathbf{c}}{\mathbf{2(M-k) - \frac{2P_r}{P_e} - \frac{k\epsilon}{P_e}}}$

This condition is generally satisfied in practical scenario. For example we can plug in our own experimental values from cifar10 and Yahoo!

$M = 5, k = 2, c = 10, R = 2000\ P_e \gg P_r, P_e \gg \epsilon$.

In this case it is easy to see that this inequality is indeed satisfied, showing us that the rate of change of the communication cost of FedMix, as a function of the number of clients will remain bounded below by DDOME.

**Conclusion**. Across all scaling dimensions of interest—experts (M), rounds (R), and clients (S)—DDOME communication is fundamentally more efficient and scalable.

## L  Missing Proof from Section 3

Here, we first state the assumption of the theorem in Section 3.

**Assumption 1.** *Let $\mathbf{\Sigma} \in \mathbb{R}^{d \times d}$ be a positive definite matrix. There exists a constant $c_{sc} > 0$ such that*

$$\lambda_{\min}\left(\mathbb{E}_{\mathbf{x} \sim \mathcal{N}(\mathbf{0}, \mathbf{\Sigma})}\left[\mathbb{I}\left\{\mathbf{x}[1] \geq \max_{\ell \in [d]} \mathbf{x}[\ell]\right\} \mathbf{x}\mathbf{x}^\top\right]\right) \geq \frac{c_{sc}}{d} \cdot \lambda_{\min}(\mathbf{\Sigma})$$

$$\mathbb{E}_{\mathbf{x} \sim \mathcal{N}(\mathbf{0}, \mathbf{\Sigma})}\left[\mathbb{I}\left\{\mathbf{x}[1] \geq \max_{\ell \in [d]} \mathbf{x}[\ell]\right\}\right] \geq \frac{c_{sc}}{d}$$

For the simplicity of the analysis, we define

$$I_j(\mathbf{x}) = \mathbb{I}\{\mathbf{x} \in \mathcal{C}_j\}; \quad I_j^\star(\mathbf{x}) = \mathbb{I}\{\mathbf{x} \in \mathcal{C}_j\}$$

### L.1 Analysis of the Initialized Router Weights

Let $\mathbf{v}_1, \ldots, \mathbf{v}_m \sim \mathcal{N}\left(\mathbf{0}, \frac{1}{d}\mathbf{I}_d\right)$. Write $\mathbf{V} = [\mathbf{v}_1, \ldots, \mathbf{v}_m] \in \mathbb{R}^{d \times m}$. By standard concentration of covariance matrix, when $m \ll d$, we have that with probability at least $1 - \exp(-\Theta(d))$

$$\left\|\mathbf{V}^\top \mathbf{V} - \mathbf{I}_m\right\|_2 \leq \frac{\sqrt{m}}{2\sqrt{d}}$$

This gives that $\sigma_m(\mathbf{V}) \geq \frac{1}{\sqrt{2}}$. For the simplicity of the analysis, we assume that this event happens. Let the singular value decomposition of $\mathbf{V}$ be given by $\mathbf{V} = \mathbf{U_V}\mathbf{\Sigma_V}\mathbf{R_V}^\top$.

**Lemma 1.** *Let $\mathbf{u}_1^\star, \ldots, \mathbf{u}_{m'}^\star$ be a fixed orthonormal list, and let $\mathbf{U}^\star = [\mathbf{u}_1^\star, \ldots, \mathbf{u}_m^\star]$ with some $m' \leq d$. Then with probability at least $1 - \exp\left(-\Theta\left(m^2 m'^2\right)\right)$ we have that*

$$\left\|\mathbf{U}^\top \mathbf{U}^\star\right\|_F \leq 2\sqrt{\frac{mm'}{d}}$$

*Proof.* Since $\mathbf{v}_j \in \mathcal{N}\left(\mathbf{0}, \frac{1}{d}\mathbf{I}_d\right)$ for all $j \in [m]$, we have that $\mathbf{U}^{\star\top}\mathbf{v}_j =: \mathbf{a}_j \sim \mathcal{N}\left(\mathbf{0}, \frac{1}{d}\mathbf{I}_{m'}\right)$. Therefore

$$\left\|\mathbf{V}^\top \mathbf{U}^\star\right\|_F^2 = \sum_{j=1}^m \|\mathbf{a}_j\|_2^2 = \sum_{i=1}^{m'} \sum_{j=1}^m \mathbf{a}_j[i]^2$$

Therefore

$$\mathbb{E}\left[\left\|\mathbf{V}^\top \mathbf{U}^\star\right\|_F^2\right] = \frac{mm'}{d}; \quad d\left\|\mathbf{U}^\top \mathbf{U}^\star\right\|_F^2 \in \mathrm{subE}\left(\Theta(mm'), \Theta(1)\right)$$

By the standard concentration inequality of Gaussian random vector, we have that

$$\Pr\left(\left\|\mathbf{V}^\top \mathbf{U}^\star\right\|_F^2 - \frac{mm'}{d} \geq \frac{mm'}{d}\right) = \Pr\left(d\left\|\mathbf{V}^\top \mathbf{U}^\star\right\|_F^2 - mm' \geq mm'\right) \leq 1 - \exp\left(-\Theta\left(m^2 m'^2\right)\right)$$

Thus, with probability at least $1 - \exp\left(-\Theta\left(m^2 m'^2\right)\right)$ we have that

$$\left\|\mathbf{V}^\top \mathbf{U}^\star\right\|_F \leq \frac{\sqrt{2mm's}}{\sqrt{d}}$$

Recall that $\mathbf{V} = \mathbf{U_V}\mathbf{\Sigma_V}\mathbf{R_V}^\top$. Therefore

$$\left\|\mathbf{V}^\top \mathbf{U}^\star\right\|_F = \left\|\mathbf{R_V}\mathbf{\Sigma_V}\mathbf{U_V}^\top \mathbf{U}^\star\right\|_F \geq \lambda_{\min}(\mathbf{V})\left\|\mathbf{U_V}^\top \mathbf{U}^\star\right\| \geq \frac{1}{\sqrt{2}}\left\|\mathbf{U_V}^\top \mathbf{U}^\star\right\|$$

Thus, we have that

$$\left\|\mathbf{U_V}^\top \mathbf{U}^\star\right\|_F \leq 2\sqrt{\frac{mm'}{d}}$$

$\square$

Recall that the ground-truth router parameters $\mathbf{v}_1^\star, \ldots, \mathbf{v}_m^\star$ is an orthonormal list, and the concatenation is denoted by $\mathbf{V}^\star$, and that $\mathbf{U_V}$ denotes the left singular vectors of $\mathbf{V}$. By Lemma 1, we have that

$$\left\|\mathbf{U_V}^\top \mathbf{V}^\star\right\|_F \leq 2 \cdot \frac{m}{\sqrt{d}} < 1$$

when $m \leq \frac{\sqrt{d}}{2}$. Therefore, $\dim\left(\mathrm{span}\left(\mathbf{U_V}\right) + \mathrm{span}\left(\mathbf{V}^\star\right)\right) = 2m$. Otherwise, if $\dim\left(\mathrm{span}\left(\mathbf{U_V}\right) + \mathrm{span}\left(\mathbf{V}^\star\right)\right) < 2m$, there exists $\mathbf{x} \in \mathrm{span}\left(\mathbf{U_V}\right) \cap \mathrm{span}\left(\mathbf{V}^\star\right)$, which implies that there exists $\mathbf{s}, \mathbf{s}' \in \mathbb{R}^m$ such that $\mathbf{U_V}\mathbf{s} = \mathbf{V}^\star\mathbf{s}'$. This gives

$$\|\mathbf{s}'\|_2 = \|\mathbf{x}\|_2 = \|\mathbf{s}\|_2 = \left\|\mathbf{U_V}^\top\mathbf{V}^\star\mathbf{s}'\right\|_2 < \|\mathbf{s}'\|_2$$

which is a contradiction. Therefore, we can let $\hat{\mathbf{U}}_\mathbf{V} \in \mathbb{R}^{d \times 2m}$ to be the matrix representing an orthonormal basis of $\mathrm{span}\left(\mathbf{U_V}\right) + \mathrm{span}\left(\mathbf{V}^\star\right)$ where the top $m$ columns are $\mathbf{U_V}$. Let $\tilde{\mathbf{U}}_\mathbf{V} \in \mathbb{R}^{d \times (d-2m)}$ be the orthogonal complement of $\hat{\mathbf{U}}_\mathbf{V}$. Then we can re-write $\mathbf{x} \sim \mathcal{N}\left(\mathbf{0}, \frac{1}{d}\mathbf{I}_d\right)$ as

$$\mathbf{x} = \hat{\mathbf{U}}_\mathbf{V}\mathbf{s}_1 + \tilde{\mathbf{U}}_\mathbf{V}\mathbf{s}_2; \quad \mathbf{s}_1 \sim \mathcal{N}\left(\mathbf{0}, \frac{1}{d}\mathbf{I}_{2m}\right); \; \mathbf{s}_2 \sim \mathcal{N}\left(\mathbf{0}, \frac{1}{d}\mathbf{I}_{d-2m}\right)$$

We make the following assumption:

**Assumption 2.** *Let $\mathbf{\Sigma} \in \mathbb{R}^{d \times d}$ be a positive definite matrix. There exists a constant $c_{sc} > 0$ such that*

$$\lambda_{\min}\left(\mathbb{E}_{\mathbf{x} \sim \mathcal{N}(\mathbf{0},\mathbf{\Sigma})}\left[\mathbb{I}\left\{\mathbf{x}[1] \geq \max_{\ell \in [d]}\mathbf{x}[\ell]\right\}\mathbf{x}\mathbf{x}^\top\right]\right) \geq \frac{c_{sc}}{d} \cdot \lambda_{\min}\left(\mathbf{\Sigma}\right)$$

$$\mathbb{E}_{\mathbf{x} \sim \mathcal{N}(\mathbf{0},\mathbf{\Sigma})}\left[\mathbb{I}\left\{\mathbf{x}[1] \geq \max_{\ell \in [d]}\mathbf{x}[\ell]\right\}\right] \geq \frac{c_{sc}}{d}$$

Let $c_{2m}$ denote the lower bound in Assumption 2 with Gaussian measure $\frac{1}{2m}$.

**Lemma 2.** *Let $\hat{\mathbf{U}}_\mathbf{V}, \tilde{\mathbf{U}}_\mathbf{V}, \mathbf{s}_1, \mathbf{s}_2$ be defined as above. If Assumption 2 hold, then we have that*

$$\mathbb{E}_{\mathbf{s}_1}\left[I_j\left(\mathbf{x}\right)\right] \geq \Omega\left(\frac{1}{m}\right); \quad \lambda_{\min}\left(\mathbb{E}_{\mathbf{s}_1}\left[I_j\left(\mathbf{x}\right)\mathbf{s}_1\mathbf{s}_1^\top\right]\right) \geq \Omega\left(\frac{1}{md}\right)$$

*Proof.* Write $\mathbf{s}_1 = \left[\mathbf{q}_1^\top, \mathbf{q}_2^\top\right]^\top$ where $\mathbf{q}_1, \mathbf{q}_2 \in \mathbb{R}^m$. Then we have that $\mathbf{q}_1, \mathbf{q}_2 \sim \mathcal{N}\left(\mathbf{0}, \frac{1}{d}\mathbf{I}_m\right)$. Moreover, by the construction of $\hat{\mathbf{U}}_\mathbf{V}$, we have that $\mathbf{q}_2$ is independent of $\mathbf{v}_1, \ldots, \mathbf{v}_m$ and $\mathbf{q}_1 = \mathbf{U_V}^\top\mathbf{x}$. Therefore

$$\mathbb{E}_{\mathbf{s}_1}\left[I_j\left(\mathbf{x}\right)\mathbf{s}_1\mathbf{s}_1^\top\right] = \mathbf{P}_1\mathbb{E}_{\mathbf{q}_1}\left[I_j\left(\mathbf{x}\right)\mathbf{q}_1\mathbf{q}_1^\top\right]\mathbf{P}_1^\top + \mathbb{E}_{\mathbf{q}_1}\left[I_j\left(\mathbf{x}\right)\right] \cdot \mathbf{P}_2\mathbb{E}_{\mathbf{q}_2}\left[\mathbf{q}_2\mathbf{q}_2^\top\right]\mathbf{P}_2^\top$$

where $\mathbf{P}_1 = \begin{bmatrix}\mathbf{I}_m \\ \mathbf{0}\end{bmatrix}$ and $\mathbf{P}_2 = \begin{bmatrix}\mathbf{0} \\ \mathbf{I}_m\end{bmatrix}$. Recall that $\mathbf{q}_1 = \mathbf{U_V}^\top\mathbf{x}$. Then we have that $\mathbf{x}^\top\mathbf{v}_j = \mathbf{q}_1^\top\mathbf{U_V}^\top\mathbf{v}_j$. Since $\mathbf{V}$ has the SVD

$$\mathbf{V} = \mathbf{U_V}^\top\mathbf{\Sigma_V}\mathbf{R_V}^\top$$

we have that $\mathbf{U_V}^\top\mathbf{v}_j = \mathbf{\Sigma_V}\mathbf{r}_j$ where $\mathbf{r}_j$ is the $j$th row of $\mathbf{R_V}$. Thus

$$\mathbf{x}^\top\mathbf{v}_j = \mathbf{q}_1^\top\mathbf{\Sigma_V}\mathbf{r}_j = \mathbf{q}_1\mathbf{\Sigma_V}\mathbf{R_V}^\top\mathbf{e}_j$$

Let $\hat{\mathbf{q}}_1 = \mathbf{R_V}\mathbf{\Sigma_V}\mathbf{q}_1$. Then we have that $\mathbf{q}_1 = \mathbf{\Sigma_V}^{-1}\mathbf{R_V}^\top\hat{\mathbf{q}}_1$, and

$$\mathbb{E}\left[\hat{\mathbf{q}}_1\hat{\mathbf{q}}_1^\top\right] = \mathbb{E}\left[\mathbf{R_V}\mathbf{\Sigma_V}\mathbf{q}_1\mathbf{q}_1^\top\mathbf{\Sigma_V}\mathbf{R_V}^\top\right] = \frac{1}{d}\mathbf{R_V}\mathbf{\Sigma_V}^2\mathbf{R_V}^\top$$

Moreover, the indicator function can thus be written as

$$I_j\left(\mathbf{x}\right) = \mathbb{I}\left\{\hat{q}_1^\top\mathbf{e}_j \geq \max_\ell \hat{q}_1^\top\mathbf{e}_\ell\right\} = \mathbb{I}\left\{\hat{\mathbf{q}}_1[j] \geq \|\hat{\mathbf{q}}_1\|_\infty\right\}$$

Thus, the original objective is re-written as

$$\mathbb{E}_{\mathbf{s}_1}\left[I_j\left(\mathbf{x}\right)\mathbf{s}_1\mathbf{s}_1^\top\right] = \mathbf{P}_1\mathbf{\Sigma_V}^{-1}\mathbf{R_V}^\top\mathbb{E}_{\hat{\mathbf{q}}_1}\left[\mathbb{I}\left\{\hat{\mathbf{q}}_1[j] \geq \|\hat{\mathbf{q}}_1\|_\infty\right\}\hat{\mathbf{q}}_1\hat{\mathbf{q}}_1^\top\right]\mathbf{R_V}\mathbf{\Sigma_V}^{-1}\mathbf{P}_1^\top$$

$$+ \frac{1}{d}\mathbb{E}_{\hat{\mathbf{q}}_1}\left[\mathbb{I}\left\{\hat{\mathbf{q}}_1[j] \geq \|\hat{\mathbf{q}}_1\|_\infty\right\}\right]\mathbf{P}_2\mathbf{P}_2^\top$$

By Assumption 2, we have that

$$\mathbb{E}_{\hat{\mathbf{q}}_1}\left[\mathbb{I}\left\{\hat{\mathbf{q}}_1[j] \geq \|\hat{\mathbf{q}}_1\|_\infty\right\}\right] \geq \Omega\left(\frac{1}{m}\right); \quad \lambda_{\min}\left(\mathbb{E}_{\hat{\mathbf{q}}_1}\left[\mathbb{I}\left\{\hat{\mathbf{q}}_1[j] \geq \|\hat{\mathbf{q}}_1\|_\infty\right\}\hat{\mathbf{q}}_1\hat{\mathbf{q}}_1^\top\right]\right) \geq \Omega\left(\frac{1}{md}\right)$$

Therefore, we have that

$$\mathbb{E}_{\mathbf{x}}\left(I_j\left(\mathbf{x}\right)\right) \geq \Omega\left(\frac{1}{m}\right); \quad \lambda_{\min}\left(\mathbb{E}_{\mathbf{s}_1}\left[I_j\left(\mathbf{x}\right)\mathbf{s}_1\mathbf{s}_1^\top\right]\right) \geq \Omega\left(\frac{1}{md}\right)$$

$\square$

*Proof of Theorem 1.* Notice that the objective in (**??**) is a quadratic form with Hessian $\mathbb{E}_{\mathbf{x}}\left[I_j\left(\mathbf{x}\right)\mathbf{x}\mathbf{x}^\top\right]$. Recall that $\mathbf{x} = \hat{\mathbf{U}}_{\mathbf{V}}\mathbf{s}_1 + \tilde{\mathbf{U}}_{\mathbf{V}}\mathbf{s}_2$. Thus, we can write

$$\mathbb{E}_{\mathbf{x}}\left[I_j\left(\mathbf{x}\right)\mathbf{x}\mathbf{x}^\top\right] = \hat{\mathbf{U}}_{\mathbf{V}}\mathbb{E}_{\mathbf{s}_1}\left[I_j\left(\mathbf{x}\right)\mathbf{s}_1\mathbf{s}_1^\top\right]\hat{\mathbf{U}}_{\mathbf{V}}^\top + \mathbb{E}_{\mathbf{s}_1}\left[I_j\left(\mathbf{x}\right)\right]\tilde{\mathbf{U}}_{\mathbf{V}}\tilde{\mathbf{U}}_{\mathbf{V}}^\top$$

since $\mathbf{s}_2$ is independent of $I_j\left(\mathbf{x}\right)$. Based on Lemma 2, we have that the objective in (**??**) is a strongly convex quadratic since the two expectation terms are strictly positive. Therefore, $\hat{\mathbf{w}}_j$ has the form

$$\hat{\mathbf{w}}_j = \left(\hat{\mathbf{U}}_{\mathbf{V}}\mathbb{E}_{\mathbf{s}_1}\left[I_j\left(\mathbf{x}\right)\mathbf{s}_1\mathbf{s}_1^\top\right]^{-1}\hat{\mathbf{U}}_{\mathbf{V}}^\top + \mathbb{E}_{\mathbf{s}_1}\left[I_j\left(\mathbf{x}\right)\right]^{-1}\tilde{\mathbf{U}}_{\mathbf{V}}\tilde{\mathbf{U}}_{\mathbf{V}}^\top\right) \cdot \mathbb{E}_{\mathbf{x}}\left[I_j\left(\mathbf{x}\right)\mathbf{x}y\right]$$

Notice that for $y$ we have

$$y = \sum_{\ell=1}^{m} I_\ell^\star\left(\mathbf{x}\right)\mathbf{x}^\top\mathbf{w}_\ell^\star$$

Therefore, $\hat{\mathbf{w}}_j$ can be written as

$$\hat{\mathbf{w}}_j = \left(\hat{\mathbf{U}}_{\mathbf{V}}\mathbb{E}_{\mathbf{s}_1}\left[I_j\left(\mathbf{x}\right)\mathbf{s}_1\mathbf{s}_1^\top\right]^{-1}\hat{\mathbf{U}}_{\mathbf{V}}^\top + \mathbb{E}_{\mathbf{s}_1}\left[I_j\left(\mathbf{x}\right)\right]^{-1}\tilde{\mathbf{U}}_{\mathbf{V}}\tilde{\mathbf{U}}_{\mathbf{V}}^\top\right)\sum_{\ell=1}^{m}\mathbb{E}_{\mathbf{x}}\left[I_j\left(\mathbf{x}\right)I_\ell^\star\left(\mathbf{x}\right)\mathbf{x}\mathbf{x}^\top\right]\mathbf{w}_\ell^\star$$

$$= \left(\hat{\mathbf{U}}_{\mathbf{V}}\mathbb{E}_{\mathbf{s}_1}\left[I_j\left(\mathbf{x}\right)\mathbf{s}_1\mathbf{s}_1^\top\right]^{-1}\hat{\mathbf{U}}_{\mathbf{V}}^\top + \mathbb{E}_{\mathbf{s}_1}\left[I_j\left(\mathbf{x}\right)\right]^{-1}\tilde{\mathbf{U}}_{\mathbf{V}}\tilde{\mathbf{U}}_{\mathbf{V}}^\top\right) \cdot$$

$$\sum_{\ell=1}^{m}\left(\hat{\mathbf{U}}_{\mathbf{V}}\mathbb{E}_{\mathbf{s}_1}\left[I_j\left(\mathbf{x}\right)I_\ell^\star\left(\mathbf{x}\right)\mathbf{s}_1\mathbf{s}_1\top\right]\hat{\mathbf{U}}_{\mathbf{V}}^\top + \mathbb{E}_{\mathbf{s}_1}\left[I_j\left(\mathbf{x}\right)I_\ell^\star\right]\tilde{\mathbf{U}}_{\mathbf{V}}\tilde{\mathbf{U}}_{\mathbf{V}}^\top\right)\mathbf{w}_\ell^\star$$

$$= \sum_{\ell=1}^{m}\left(\hat{\mathbf{U}}_{\mathbf{V}}\mathbb{E}_{\mathbf{s}_1}\left[I_j\left(\mathbf{x}\right)\mathbf{s}_1\mathbf{s}_1^\top\right]^{-1}\mathbb{E}_{\mathbf{s}_1}\left[I_j\left(\mathbf{x}\right)I_\ell^\star\left(\mathbf{x}\right)\mathbf{s}_1\mathbf{s}_1\top\right]\hat{\mathbf{U}}_{\mathbf{V}}^\top + \mathbb{E}_{\mathbf{s}_1}\left[I_j\left(\mathbf{x}\right)\right]^{-1}\mathbb{E}_{\mathbf{s}_1}\left[I_j\left(\mathbf{x}\right)I_\ell^\star\right]\tilde{\mathbf{U}}_{\mathbf{V}}\tilde{\mathbf{U}}_{\mathbf{V}}^\top\right)\mathbf{w}_\ell^\star$$

Let $\mathbf{w}_r^\star$ be given for any $r \in [m]$. Then we have that

$$\mathbf{w}_r^\star = \hat{\mathbf{U}}_{\mathbf{V}}\hat{\mathbf{U}}_{\mathbf{V}}^\top + \tilde{\mathbf{U}}_{\mathbf{V}}\tilde{\mathbf{U}}_{\mathbf{V}}^\top$$

Therefore, the difference between $\hat{\mathbf{w}}_j$ and $\mathbf{w}_r^\star$ is given by

$$\hat{\mathbf{w}}_j - \mathbf{w}_r^\star = \tilde{\mathbf{U}}_{\mathbf{V}}\left(\sum_{\ell=1}^{m}\mathbb{E}_{\mathbf{s}_1}\left[I_j\left(\mathbf{x}\right)\right]^{-1}\mathbb{E}_{\mathbf{s}_1}\left[I_j\left(\mathbf{x}\right)I_\ell^\star\left(\mathbf{x}\right)\right]\tilde{\mathbf{U}}_{\mathbf{V}}^\top\mathbf{w}_\ell^\star\right) - \tilde{\mathbf{U}}_{\mathbf{V}}\tilde{\mathbf{U}}_{\mathbf{V}}^\top\mathbf{w}_r^\star$$

$$+ \hat{\mathbf{U}}_{\mathbf{V}}\sum_{\ell=1}^{m}\mathbb{E}_{\mathbf{s}_1}\left[I_j\left(\mathbf{x}\right)\mathbf{s}_1\mathbf{s}_1^\top\right]^{-1}\mathbb{E}_{\mathbf{s}_1}\left[I_j\left(\mathbf{x}\right)I_\ell^\star\left(\mathbf{x}\right)\mathbf{s}_1\mathbf{s}_1\top\right]\hat{\mathbf{U}}_{\mathbf{V}}^\top\mathbf{w}_\ell^\star - \hat{\mathbf{U}}_{\mathbf{V}}\hat{\mathbf{U}}_{\mathbf{V}}^\top\mathbf{w}_r^\star$$

Due to the orthogonality between $\hat{\mathbf{U}}_{\mathbf{V}}$ and $\tilde{\mathbf{U}}_{\mathbf{V}}$, the magnitude of $\hat{\mathbf{w}}_j - \mathbf{w}_r^\star$ must be lower bounded by the magnitude of its projection onto $\tilde{\mathbf{U}}_{\mathbf{V}}$. Thus

$$\|\hat{\mathbf{w}}_j - \mathbf{w}_r^\star\|_2 \geq \left\|\tilde{\mathbf{U}}_{\mathbf{V}}\tilde{\mathbf{U}}_{\mathbf{V}}^\top\left(\sum_{\ell=1}^{m}\mathbb{E}_{\mathbf{s}_1}\left[I_j\left(\mathbf{x}\right)\right]^{-1}\mathbb{E}_{\mathbf{s}_1}\left[I_j\left(\mathbf{x}\right)I_\ell^\star\left(\mathbf{x}\right)\right]\mathbf{w}_\ell^\star - \mathbf{w}_r^\star\right)\right\|_2$$

$$\geq \left\|\sum_{\ell=1}^{m}\mathbb{E}_{\mathbf{s}_1}\left[I_j\left(\mathbf{x}\right)\right]^{-1}\mathbb{E}_{\mathbf{s}_1}\left[I_j\left(\mathbf{x}\right)I_\ell^\star\left(\mathbf{x}\right)\right]\mathbf{w}_\ell^\star - \mathbf{w}_r^\star\right\|_2$$

$$- \left\|\left(\mathbf{I}_d - \tilde{\mathbf{U}}_{\mathbf{V}}\tilde{\mathbf{U}}_{\mathbf{V}}^\top\right)\left(\sum_{\ell=1}^{m}\mathbb{E}_{\mathbf{s}_1}\left[I_j\left(\mathbf{x}\right)\right]^{-1}\mathbb{E}_{\mathbf{s}_1}\left[I_j\left(\mathbf{x}\right)I_\ell^\star\left(\mathbf{x}\right)\right]\mathbf{w}_\ell^\star - \mathbf{w}_r^\star\right)\right\|_2$$

Similarly, we can obtain that

$$\left\|\tilde{\mathbf{U}}_{\mathbf{V}}^\top\left(\hat{\mathbf{w}}_j-\mathbf{w}_r^\star\right)\right\|_2 \geq \left\|\tilde{\mathbf{U}}_{\mathbf{V}}\tilde{\mathbf{U}}_{\mathbf{V}}^\top\left(\sum_{\ell=1}^m \mathbb{E}_{\mathbf{s}_1}\left[I_j\left(\mathbf{x}\right)\right]^{-1}\mathbb{E}_{\mathbf{s}_1}\left[I_j\left(\mathbf{x}\right)I_\ell^\star\left(\mathbf{x}\right)\right]\mathbf{w}_\ell^\star-\mathbf{w}_r^\star\right)\right\|_2$$

$$\geq \left\|\sum_{\ell=1}^m \mathbb{E}_{\mathbf{s}_1}\left[I_j\left(\mathbf{x}\right)\right]^{-1}\mathbb{E}_{\mathbf{s}_1}\left[I_j\left(\mathbf{x}\right)I_\ell^\star\left(\mathbf{x}\right)\right]\mathbf{w}_\ell^\star-\mathbf{w}_r^\star\right\|_2$$

$$-\left\|\left(\mathbf{I}_d-\tilde{\mathbf{U}}_{\mathbf{V}}\tilde{\mathbf{U}}_{\mathbf{V}}^\top\right)\left(\sum_{\ell=1}^m \mathbb{E}_{\mathbf{s}_1}\left[I_j\left(\mathbf{x}\right)\right]^{-1}\mathbb{E}_{\mathbf{s}_1}\left[I_j\left(\mathbf{x}\right)I_\ell^\star\left(\mathbf{x}\right)\right]\mathbf{w}_\ell^\star-\mathbf{w}_r^\star\right)\right\|_2$$

We further notice that

$$\sum_{\ell=1}^m \mathbb{E}_{\mathbf{s}_1}\left[I_j\left(\mathbf{x}\right)\right]^{-1}\mathbb{E}_{\mathbf{s}_1}\left[I_j\left(\mathbf{x}\right)I_\ell^\star\left(\mathbf{x}\right)\right]\mathbf{w}_\ell^\star-\mathbf{w}_r^\star = \mathbf{W}^\star\boldsymbol{\zeta}_j;\ \boldsymbol{\zeta}_j[\ell]=\mathbb{E}_{\mathbf{s}_1}\left[I_j\left(\mathbf{x}\right)\right]^{-1}\mathbb{E}_{\mathbf{s}_1}\left[I_j\left(\mathbf{x}\right)I_\ell^\star\left(\mathbf{x}\right)\right]-\mathbb{I}\left\{j=\ell\right\}$$

Therefore, we have that

$$\left\|\hat{\mathbf{w}}_j-\mathbf{w}_r^\star\right\|_2 \geq \left\|\tilde{\mathbf{U}}_{\mathbf{V}}\tilde{\mathbf{U}}_{\mathbf{V}}^\top\mathbf{W}^\star\boldsymbol{\zeta}_j\right\|_2 = \left\|\tilde{\mathbf{U}}_{\mathbf{V}}^\top\mathbf{W}^\star\boldsymbol{\zeta}_j\right\| \geq \left(1-\left\|\hat{\mathbf{U}}_{\mathbf{V}}^\top\mathbf{W}^\star\right\|_2\right)\left\|\boldsymbol{\zeta}_j\right\|_2$$

By Lemma 1, we have that with probability at least $1-\exp\left(-\Theta\left(m^4\right)\right)$, we have that $\left\|\hat{\mathbf{U}}_{\mathbf{V}}^\top\mathbf{W}^\star\right\|_2 \leq$ $\frac{3m}{\sqrt{d}}$. Moreover, diving deeper into $\left\|\boldsymbol{\zeta}\right\|_j$, we have

$$\left\|\boldsymbol{\zeta}_j\right\|_2^2 = \sum_{\ell=1}^m \left(\mathbb{E}_{\mathbf{s}_1}\left[I_j\left(\mathbf{x}\right)\right]^{-1}\mathbb{E}_{\mathbf{s}_1}\left[I_j\left(\mathbf{x}\right)I_\ell^\star\left(\mathbf{x}\right)\right]-\mathbb{I}\left\{j=\ell\right\}\right)^2$$

$$= \sum_{\ell\neq r}^m \mathbb{E}_{\mathbf{s}_1}\left[I_j\left(\mathbf{x}\right)\right]^{-2}\mathbb{E}_{\mathbf{s}_1}\left[I_j\left(\mathbf{x}\right)I_\ell^\star\left(\mathbf{x}\right)\right]^2 + \left(E_{\mathbf{s}_1}\left[I_j\left(\mathbf{x}\right)\right]^{-1}\mathbb{E}_{\mathbf{s}_1}\left[I_j\left(\mathbf{x}\right)I_\ell^\star\left(\mathbf{x}\right)\right]-1\right)^2$$

$$\geq \sum_{\ell\neq r}\mathbb{E}_{\mathbf{s}_1}\left[I_j\left(\mathbf{x}\right)I_\ell^\star\left(\mathbf{x}\right)\right]^2 + \left(E_{\mathbf{s}_1}\left[I_j\left(\mathbf{x}\right)\right]^{-1}\mathbb{E}_{\mathbf{s}_1}\left[I_j\left(\mathbf{x}\right)I_\ell^\star\left(\mathbf{x}\right)\right]-1\right)^2$$

Since $E_{\mathbf{s}_1}\left[I_j\left(\mathbf{x}\right)\right]^{-1}\mathbb{E}_{\mathbf{s}_1}\left[I_j\left(\mathbf{x}\right)I_\ell^\star\left(\mathbf{x}\right)\right] \leq 1$, and by Lemma 2, we have

$$E_{\mathbf{s}_1}\left[I_j\left(\mathbf{x}\right)\right]^{-1} \geq \Omega\left(\frac{1}{m}\right)$$

Therefore, we can obtain that

$$\left(E_{\mathbf{s}_1}\left[I_j\left(\mathbf{x}\right)\right]^{-1}\mathbb{E}_{\mathbf{s}_1}\left[I_j\left(\mathbf{x}\right)I_\ell^\star\left(\mathbf{x}\right)\right]-1\right)^2 \geq \left(\frac{m}{c}\mathbb{E}_{\mathbf{s}_1}\left[I_j\left(\mathbf{x}\right)I_\ell^\star\left(\mathbf{x}\right)\right]-1\right)^2$$

for some constant $c$. This gives that

$$\left\|\boldsymbol{\zeta}_j\right\|_2^2 \leq \sum_{\ell\neq r}\mathbb{E}_{\mathbf{s}_1}\left[I_j\left(\mathbf{x}\right)I_\ell^\star\left(\mathbf{x}\right)\right]^2 + \left(\frac{m}{c}\mathbb{E}_{\mathbf{s}_1}\left[I_j\left(\mathbf{x}\right)I_\ell^\star\left(\mathbf{x}\right)\right]-1\right)^2 \leq \left\|\hat{\boldsymbol{\zeta}}_j\right\|_2^2$$

where

$$\hat{\boldsymbol{\zeta}}_j[\ell] = \begin{cases} \mathbb{E}_{\mathbf{s}_1}\left[I_j\left(\mathbf{x}\right)I_\ell^\star\left(\mathbf{x}\right)\right] & \text{if } \ell\neq j \\ \frac{m}{c}\mathbb{E}_{\mathbf{s}_1}\left[I_j\left(\mathbf{x}\right)I_\ell^\star\left(\mathbf{x}\right)\right]-1 & \text{if } \ell=j \end{cases}$$

This gives

$$\mathbb{E}_{\mathbf{V}}\left[\hat{\boldsymbol{\zeta}}_j\right][\ell] = \begin{cases} \frac{1}{m^2} & \text{if } \ell\neq j \\ \frac{1}{mc}-1 & \text{if } \ell=j \end{cases}$$

Therefore,

$$\mathbb{E}_{\mathbf{V}}\left[\left\|\hat{\mathbf{w}}_j-\mathbf{w}_r^\star\right\|_2\right] \geq \mathbb{E}_{\mathbf{V}}\left\|\hat{\boldsymbol{\zeta}}_j\right\|_2 \geq \left\|\mathbb{E}_{\mathbf{V}}\left[\hat{\boldsymbol{\zeta}}_j\right]\right\|_2 = \left(\frac{1}{m^3}+\left(1-\frac{1}{mc}\right)\right)^2 = \Omega\left(1\right)$$

Using a similar approach, we can conclude that

$$\mathbb{E}_{\mathbf{V}} \left[ \left\| \tilde{\mathbf{U}}_{\mathbf{V}}^{\top} \left( \hat{\mathbf{w}}_j - \mathbf{w}_r^{\star} \right) \right\|_2 \right] \geq \Omega \left( 1 \right)$$

Therefore, for the test loss, we can write

$$
\begin{aligned}
\mathcal{L} \left( \hat{\boldsymbol{\theta}} \right) &= \frac{1}{2} \mathbb{E}_{\mathbf{x},y} \left[ \sum_{j=1}^{m} I_j \left( \mathbf{x} \right) \left( \mathbf{w}_j^{\top} \mathbf{x} - y \right)^2 \right] \\
&= \frac{1}{2} \mathbb{E}_{\mathbf{x},y} \left[ \sum_{j=1}^{m} I_j \left( \mathbf{x} \right) \left( \mathbf{w}_j^{\top} \mathbf{x} - \sum_{j'=1}^{m} I_{j'}^{\star} \left( \mathbf{x} \right) \mathbf{w}_{j'}^{\star \top} \mathbf{x} \right)^2 \right] \\
&= \frac{1}{2} \sum_{j,j'=1}^{m} \left( \mathbf{w}_j - \mathbf{w}_{j'}^{\star} \right)^{\top} \mathbb{E}_{\mathbf{x}} \left[ I_j \left( \mathbf{x} \right) I_{j'}^{\star} \left( \mathbf{x} \right) \mathbf{x} \mathbf{x}^{\top} \right] \left( \mathbf{w}_j - \mathbf{w}_{j'}^{\star} \right) \\
&= \frac{1}{2} \sum_{j,j'=1}^{m} \left( \mathbf{w}_j - \mathbf{w}_{j'}^{\star} \right)^{\top} \left( \mathbb{E}_{\mathbf{x}} \left[ I_j \left( \mathbf{x} \right) I_{j'}^{\star} \left( \mathbf{x} \right) \mathbf{s}_1 \mathbf{s}_1^{\top} \right] + \mathbb{E}_{\mathbf{s}_1} \left[ I_j \left( \mathbf{x} \right) I_{j'}^{\star} \left( \mathbf{x} \right) \right] \tilde{\mathbf{U}}_{\mathbf{V}} \tilde{\mathbf{U}}_{\mathbf{V}}^{\top} \right) \left( \mathbf{w}_j - \mathbf{w}_{j'}^{\star} \right) \\
&\geq \frac{1}{2} \sum_{j,j'=1}^{m} \mathbb{E}_{\mathbf{s}_1} \left[ I_j \left( \mathbf{x} \right) I_{j'}^{\star} \left( \mathbf{x} \right) \right] \cdot \left( \mathbf{w}_j - \mathbf{w}_{j'}^{\star} \right)^{\top} \tilde{\mathbf{U}}_{\mathbf{V}} \tilde{\mathbf{U}}_{\mathbf{V}}^{\top} \left( \mathbf{w}_j - \mathbf{w}_{j'}^{\star} \right) \\
&\geq \frac{1}{2} \left\| \tilde{\mathbf{U}}_{\mathbf{V}} \left( \mathbf{w}_j - \mathbf{w}_j^{\star} \right) \right\|_2^2
\end{aligned}
$$

This gives that

$$\mathbb{E}_{\mathbf{V}} \left[ \mathcal{L} \left( \hat{\boldsymbol{\theta}} \right) \right] \geq \frac{1}{2} \mathbb{E} \left[ \left\| \tilde{\mathbf{U}}_{\mathbf{V}} \left( \mathbf{w}_j - \mathbf{w}_j^{\star} \right) \right\|_2^2 \right] \geq \frac{1}{2} \mathbb{E}_{\mathbf{V}} \left[ \left\| \hat{\mathbf{w}}_j - \mathbf{w}_r^{\star} \right\|_2 \right]^2 \geq \Omega \left( 1 \right)$$

$\square$

