# OpenReview forum: "Learning to Specialize: Joint Gating-Expert Training for Adaptive MoEs in Decentralized Settings"
_NeurIPS.cc/2025/Conference — NeurIPS 2025 poster_

### Official Review · Reviewer_tKQk · 2025-06-14

**Clarity:** 4
**Significance:** 2
**Originality:** 2
**Rating:** 4
**Confidence:** 3

**Summary:**

The authors study the issue of expert specialization in an MoE framework (specifically in the context of federated learning). They argue through theoretical insights in a Top-1 setting with linear experts that joint training of experts’ weights and their gating function is necessary, else the loss is lower-bounded by a constant. The authors then propose a dynamic expert allocation framework to use implicit client information to achieve specialization without the need for explicit labels (unlike prior work). The authors perform experiments on a range of both image and text classification tasks (with thorough ablation studies), with SOTA performance on zero-shot personalization scores.

**Questions:**

- **[Q1]**: On [L79], should the local loss function be $\ell(\mathbf{W}_s,\mathcal{D}_s)$ (i.e., with the subscript)? Might the authors clarify whether is a typo, or instead further elaborate on the distinction they intend to draw here between this and the definitions of the partial loss terms above it?

- **[Q2]** In the “remarks” on Theorem 1, the authors state the implication of the loss lower bound when updating the *experts first*, followed by the weights, and then conclude with the importance of joint training. In my impression, it’s important that the result also holds when the update is performed in the reverse order. I think this might follow trivially from the authors’ theorem, but i believe it’s important to spell that out explicitly. Might the authors have comments on whether this holds, or instead perhaps: to what extent this is not important?

- **[Q3]** Is there any important reason why the authors name the shared branch an “expert” (e.g., [L161])? I think this could be confused with the recent design trend of using a shared expert [1] (that runs in parallel with the conditional ones, rather than before as a standalone feature extractor). In my current understanding, I would avoid calling this step an “expert”.

- **[Q4]** The authors state that load balance is a problem with certain choice of pre-trained backbones. This acknowledged limitation is insightful, but I wonder whether the use of a standard load balancing loss [2] might mitigate the issues experienced by the authors. Did the authors consider this at all?

---

[1]: Dai, Damai, et al. "Deepseekmoe: Towards ultimate expert specialization in mixture-of-experts language models." *arXiv preprint arXiv:2401.06066* (2024).

[2]: Fedus, William, Barret Zoph, and Noam Shazeer. "Switch transformers: Scaling to trillion parameter models with simple and efficient sparsity." *Journal of Machine Learning Research* 23.120 (2022): 1-39.

**Ethical Concerns:**

["NO or VERY MINOR ethics concerns only"]

**Final Justification:**

The paper and theoretical framework would be of interest to the research community; but I am unconvinced by the authors' rebuttal to strongly support the paper

**Limitations:**

Yes; the authors' limitations are listed. An extra comment here about the experiments’ limited scope to classification tasks would be useful however.

**Quality:**

4

**Strengths And Weaknesses:**

# Strengths

## [S1] Timely and important topic

The paper tackles some very interesting and important problems. With the popularity of MoEs, it’s of great value to study how, why, and when expert *expertise* emerges. The topic here studied by the authors is of great interest to the community.

## [S2] The paper writing is clear

The paper is very well written--the authors do a fantastic job of introducing the issues with expert specialization and its importance. Ultimately, the high quality of the writing not only makes the paper crystal clear and compelling, but also an enjoyable read.

## [S3] Theoretical framework is compelling and insightful

Whilst I do have a question regarding the scope of the result shown ([Q2] below), the theoretical framework presented with top-1 routing and linear experts is a remarkably clean and useful setting in which to motivate the authors’ argument theoretically. Whilst it is admittedly contrived (specific to top-1 routing and makes the assumption of fixed ground-truth expert assignments), I believe it constitutes an insightful framework with which future researchers could further study MoEs theoretically.

# Weaknesses

## [W1] clarifying federated learning scope

The authors do a good job of being fairly explicit in the paper’s title and abstract that the work explores issues with MoEs in the context of federated learning. My impression however is that adding additional clarification when discussing the contributions [L30-L48] about the federated learning scope would help make the work even clearer. More than this, however, the authors should elaborate on the extent to which the insights in the paper do (or do not) relate to standard MoE settings outside of the federated learning paradigm. I imagine a non-trivial percentage of readers may not be experts in the federated learning MoE setting specifically, in which case some discussion in the introduction would help the insights bridge this gap.

## [W2] experiments are limited to classification tasks

Firstly, I must state that I am not an expert on federated learning, so I understand that it might be non-trivial to adapt FL pipelines to all experimental setups. Whilst, I appreciate the authors’ efforts to perform experiments on both the vision and text domains, it would strengthen the paper even more to have even small, proof-of-concept experiments on a modern generative task. For example, the authors could (partially) train a NanoGPT (https://github.com/karpathy/nanoGPT) with the RTX A6000 used in their other experiments on a toy data dataset such as TinyStories. I appreciate there might be difficulties that i do not foresee here however, and if that is the case, a discussion of why this is hard to extend to the generative paradigm will be well-placed in the authors’ limitations section.

## [W3] Confusing notation and formatting

1. I find the number of sentences typeset in *“quotations and italics”* in the introduction to be mildly confusing (e.g. [L37]). It’s not clear to me if these are direct quotes from prior work or not--my personal suggestion here as a reader would be to remove the quotations if these are not verbatim quotes from prior work to avoid confusion.
2. I think the choice to use $\mathbf{W}_r$ for the gating function’s parameters (when $\mathbf{W}_i$ is already used to denote the *weights* of a particular expert) is confusing. Furthermore, I do not understand the need for the $_r$ subscript. Might the authors have a justification for this that I overlooked? If not, I might suggest using a totally different letter for the expert gate (e.g. $\mathbf{G}$) could prevent confusion here, without any subscript.

---

> ### Author Rebuttal · Authors · 2025-07-31
>
> We sincerely thank the reviewer for their thoughtful and encouraging feedback. We are especially grateful for the recognition of our theoretical framework, writing clarity, and the significance of the problem tackled. The reviewer’s comments on the relevance and presentation are deeply appreciated, as we invested significant effort in making the paper both accessible and compelling to the broader machine learning community. The reviewer raises several concerns regarding scope clarification, notation, and the potential for broader applicability. We address these points below.
>
>  1. **"clarifying federated learning scope:
> The authors do a good job of being fairly explicit in the paper’s title and abstract that the work explores issues with MoEs in the context of federated learning. My impression however is that adding additional clarification when discussing the contributions [L30-L48] about the federated learning scope would help make the work even clearer. More than this, however, the authors should elaborate on the extent to which the insights in the paper do (or do not) relate to standard MoE settings outside of the federated learning paradigm. I imagine a non-trivial percentage of readers may not be experts in the federated learning MoE setting specifically, in which case some discussion in the introduction would help the insights bridge this gap."**
>
>     - Thank you for the opportunity to clarify the scope of the work here. As we tried our best to frame the results and insights obtained from the study for general MoE setups, not just for FL. **_To clarify the potential misunderstanding: all the results and insights in the paper indeed apply to standard MoE settings._** The theoretical result is concerned with the dynamics of learning underlying distributions and is not specific to the federated learning setting. Furthermore, the DDOME system is not tied exclusively to FL but applies to any decentralized framework. The primary motivation of this paper is to investigate whether expert specialization can occur in a single-domain setting where explicit task boundaries are not available. From this perspective, **we ask: _what learning setup severely suffers from diverse data distributions in a single-domain scenario?_** The answer is Federated Learning. FL has long been studied not only for its practical relevance but also because of the unique challenges it presents, particularly due to the inherently non-i.i.d. nature of its data, a scenario that conventional centralized methods often struggle to handle effectively. **_In this work, we not only show that expert specialization is possible under such conditions, but also propose a general decentralized framework that, when applied to FL, yields highly competitive results._**
>
>
> 2. **"Confusing notation and formatting:**
>
>     2.1. **I find the number of sentences typeset in “quotations and italics” in the introduction to be mildly confusing (e.g. [L37]). It’s not clear to me if these are direct quotes from prior work or not--my personal suggestion here as a reader would be to remove the quotations if these are not verbatim quotes from prior work to avoid confusion.**
>
>       2.2. **I think the choice to use $W_r$ for the gating function’s parameters (when $W_i$ is already used to denote the weights of a particular expert) is confusing. Furthermore, I do not understand the need for the subscript. Might the authors have a justification for this that I overlooked? If not, I might suggest using a totally different letter for the expert gate (e.g. $G$) could prevent confusion here, without any subscript"**
>
>       - To clarify the first point (2.1.), the use of quotation and italics in the introduction was purely stylistic and intended to emphasize key statements or ideas. However, we understand that this formatting may have caused confusion but those lines are not direct quotes from prior work. As for the second point (2.2.) the notation $W_r$ is standard in some seminal MoE literature, where $r$ denotes the router. We followed this convention for consistency with prior work, **Switch Transformers: Scaling to Trillion Parameter Models with Simple and Efficient Sparsity** (Fedus et al., 2022).
>
>  3. **"experiments are limited to classification tasks
> Firstly, I must state that I am not an expert on federated learning, so I understand that it might be non-trivial to adapt FL pipelines to all experimental setups. Whilst, I appreciate the authors’ efforts to perform experiments on both the vision and text domains, it would strengthen the paper even more to have even small, proof-of-concept experiments on a modern generative task. For example, the authors could (partially) train a NanoGPT with the RTX A6000 used in their other experiments on a toy data dataset such as TinyStories. I appreciate there might be difficulties that i do not foresee here however, and if that is the case, a discussion of why this is hard to extend to the generative paradigm will be well-placed in the authors’ limitations section."**
>
>     - In principle, training specialized NanoGPT models within our framework is possible.  However, such a setup introduces significant practical and infrastructure-related challenges. The server would need to manage and store multiple NanoGPT models concurrently, while also handling the large-scale parameter updates returned from clients. Although possible in principle, given hardware and time constraints, such experiments were infeasible. **That said, a key future direction addresses this. We are exploring coordinated systems of small, diverse models to emulate LLM-like performance.** Our goal is to demonstrate that a coordinated system of diverse lightweight expert models can achieve competitive performance with larger models such as NanoGPT or Multimodel LLMs. We believe this direction holds great promise and aligns closely with the motivations behind this point.
>
> **Questions**
>
>   Q1. **"On [L79], should the local loss function be $\mathcal{L}(\mathcal{W}_s, \mathcal{D}_s)$
>  (i.e., with the subscript)? Might the authors clarify whether is a typo, or instead further elaborate on the distinction they intend to draw here between this and the definitions of the partial loss terms above it? "**
>
>    - Yes, the local loss function is $\mathcal{L}(\mathcal{W}_s, \mathcal{D}_s)$. The notation without a subscript was an intentional abstraction to represent the shared objective across both local (client) and global loss functions.
>
>   Q2. **"In the “remarks” on Theorem 1, the authors state the implication of the loss lower bound when updating the experts first, followed by the weights, and then conclude with the importance of joint training. In my impression, it’s important that the result also holds when the update is performed in the reverse order. I think this might follow trivially from the authors’ theorem, but i believe it’s important to spell that out explicitly. Might the authors have comments on whether this holds, or instead perhaps: to what extent this is not important? "**
>
>    - Thank you for asking the question. Indeed, the proof strategy for the "router-first" case would be similar to our current result. In particular, in proving Theorem 1, the core idea lies in showing that the data points assigned to train each expert is a mix of roughly equal number of data points from each cluster. To show the "router-first" case, we simply notice that a data point from cluster $j$ will be assigned to a randomly initialized expert with parameter $w_r$ if $w_r - w_j^\star$ projected onto $x$ has the smallest magnitude. Therefore, the learned router will try to minimize the projected magnitude of directions $w_r - w_j^\star$. Due to the randomness of $w_r$, the learned router will also be roughly random.
>
>  Q3. **"Is there any important reason why the authors name the shared branch an “expert” (e.g., [L161])? I think this could be confused with the recent design trend of using a shared expert [1] (that runs in parallel with the conditional ones, rather than before as a standalone feature extractor). In my current understanding, I would avoid calling this step an “expert"**
>
>   - We are assuming the reviewer is referring to the “common-expert.” On second thought, a more appropriate name could have been “common-embedder.” The original term was chosen because this component embeds the data to guide expert selection. We will consider renaming it.
>
>  Q4. **"The authors state that load balance is a problem with certain choice of pre-trained backbones. This acknowledged limitation is insightful, but I wonder whether the use of a standard load balancing loss [2] might mitigate the issues experienced by the authors. Did the authors consider this at all?"**
>
>    - Although this is an interesting direction, its implementation is not as straightforward. In standard MoEs load balancing is used to encourage uniform expert utilization across input tokens. DDOME operates under a slightly different paradigm: the router selects a topk subset of experts to be sent to each client, based on the entirety of each client's data. This expert-to-client assignment is explicitly designed to reflect the client's unique data characteristics. In this setup, enforcing uniform expert usage—i.e., ensuring all experts are equally used across clients—could undermine DDOME’s core objective of learning specialized experts aligned with client-level heterogeneity. When imbalance does arise, we do not view it as a flaw, but rather as an emergent property of specialization. To address router collapse, we introduced a fundamentally different mechanism: the anchor-client strategy. Unlike traditional load balancing, which injects uniformity through the process, our approach establishes persistent relationships between certain clients and experts to encourage stable and diverse expert specialization. We believe this novel contribution could have value even beyond FL and may inspire future work in general MoE designs.

---

> ### Comment · Reviewer_tKQk · 2025-08-01
>
> Thanks to the authors for their thorough response.
>
> I maintain that the paper is a useful addition to the research community; particularly the theoretical framework, which I think could be inspiring for future theoretical insights into MoEs.
>
> I do not find the authors' rebuttal compelling, however--the authors do acknowledge the points raised but do not commit to making any changes or amends. Given, I already weakly support the paper, I maintain my weakly positive score.

---

### Official Review · Reviewer_Uwk3 · 2025-06-29

**Clarity:** 3
**Significance:** 2
**Originality:** 3
**Rating:** 4
**Confidence:** 4

**Summary:**

This paper introduces Dynamically Decentralized Orchestration of MoEs (DDOME), a novel framework tailored for decentralized learning environments, particularly Federated Learning (FL). DDOME addresses the challenge of achieving meaningful expert specialization in single-domain tasks where explicit data partitions are absent, by leveraging implicit data heterogeneity across decentralized sources. The authors given a theoretic justification for why joint training of gating mechanisms and experts is necessary, to avoid suboptimal specialization. DDOME utilizes a pretrained common expert as a feature extractor to inform a dynamic gating function, which adaptively selects subsets of experts for just-in-time personalization on unseen clients. The framework incorporates "anchor clients" to guide early and stable expert specialization. Empirical validation within FL demonstrates DDOME's superior accuracy (4% to 24% improvement) over state-of-the-art FL baselines in image and text classification, while also reducing communication costs compared to other MoE based FL methods.

**Questions:**

- Robustness to Common Expert Quality: Given the heavy reliance on a high-quality pretrained common expert, what are the proposed strategies for DDOME's performance in scenarios where such an expert is unavailable, or its performance is sub-optimal? Have the authors considered methods for adaptive or periodic updates to the common expert itself within a federated setup to alleviate this dependency?
- Scalability to Million-Client Scale: The current experiments use 100 clients. Could the authors discuss the major practical hurdles (e.g., communication bandwidth, computational cost of gating function for very large M, synchronization overhead) and potential solutions for scaling DDOME to federated deployments involving millions of clients and a much larger pool of experts?

**Ethical Concerns:**

["NO or VERY MINOR ethics concerns only"]

**Final Justification:**

I have updated from score from 3 to 4 after the clarifications provided by the authors.

**Limitations:**

The authors have adequately addressed the limitations and potential negative societal impact of their work. They provide a dedicated "Limitations" section discussing key constraints such as the dependence on pretrained common experts, remaining communication overhead, expert initialization challenges, limited scalability testing, and complexity of gating mechanism tuning. Furthermore, a "Broader Impacts" section thoughtfully covers both positive societal impacts (e.g., privacy-preserving FL, resource efficiency) and potential negative impacts (e.g., exacerbating fairness issues due to bias amplification) while suggesting mitigation.

**Paper Formatting Concerns:**

Typos:
- Page 1, Line 14: "up to an 24% accuracy improvement" -> "up to a 24% accuracy improvement".
- Page 2, Line 94: “an ground-truth” -> “a ground-truth”
- Page 3, Line 115: “training experts first following by” -> “training experts first followed by”
- Page 4, Line 150, “sparse-enforced” -> “sparsity-enforced”
- Page 7, Line 293: "became unstable during training (10% for CIFAR10 / <5% for CIFAR100)." The percentages seem to refer to accuracy; clarifying this would be helpful (e.g., "achieving only 10% accuracy for CIFAR10").
- Page 14, Table 4: "77.8%%" contains a double percent sign. It should be "77.8%".

**Quality:**

2

**Strengths And Weaknesses:**

Strengths:
- Novelty in Decentralized MoEs: DDOME presents a novel solution for dynamic expert specialization in challenging decentralized and single-domain settings. The integration of a pretrained common expert to guide the gating function and the "anchor clients" mechanism are novel contributions.
- Theoretical Foundation: The paper provides a rigorous theoretical justification for the necessity of joint training of gating and expert modules, demonstrating that decoupled training leads to suboptimal specialization. This theoretical insight is a significant contribution that underpins the proposed methodology.
- Harder setup: Previous approaches assume access to part of the new client’s labeled data is available. This limitation is not imposed in the paper, as new users are likely unwilling/unable to provide accurate labeled data and might not have sufficient resources to contribute to a fine-tuning phase of the whole model.
- Impressive Empirical Performance: DDOME consistently achieves state-of-the-art accuracy improvements (ranging from 4% to 24%) over leading FL baselines and text classification tasks. Its performance in zero-shot personalization on unseen clients is particularly noteworthy.
- Robustness to Data Heterogeneity: The framework effectively leverages implicit client-specific data characteristics to dynamically specialize experts and performs strongly in scenarios with highly skewed data distributions (e.g., Dirichlet distribution).

Weaknesses:
- Heavy Dependence on Pretrained Common Expert: The success of DDOME is critically dependent on the availability and quality of an effective pretrained common expert. The ablation study explicitly shows that a poor common expert can lead to performance collapse. This is a significant practical limitation, as acquiring such a well-performing expert might be costly or not feasible in all decentralized scenarios.
- Small and Uninteresting datasets: Datasets considered from FL literature are too old and toyish from before 2021, and appear to be non-interesting. The authors should consider more interesting and larger datasets.
- Lack of Statistical Significance: The paper does not report error bars for its empirical results. This makes it challenging to ascertain the robustness and generalizability of the reported accuracy improvements across multiple runs and different random seeds.
- Choice of hyperparameters: The authors consider a different optimizer for DDOME than the one considered for baseline models in lines 241-243. This appears problematic without a solid justification. Perhaps the change in optimizer is the cause for some of the gains.
- Ambiguity in M=K Scenario: In experiments where the number of experts equals the number of selected experts (M=K), particularly for M=K=2 on CIFAR10, DDOME's performance (60.16%) is lower than the initial common expert's accuracy (73.39%) [Table 7]. The authors attribute this to "malicious 'interference'", which is a vague explanation. This suggests a potential fragility or a specific failure mode in DDOME under certain configurations that needs deeper investigation and mitigation.
- Relative Communication Cost: While DDOME reduces communication compared to some MoE methods like FedMix, the authors acknowledge it "still incurs higher communication overhead compared to traditional single-model methods". This is an important trade-off that should be more explicitly highlighted in the main claims regarding communication efficiency.
- Reproducibility: The paper does not provide code.

---

> ### Author Rebuttal · Authors · 2025-07-30
>
> We thank the reviewer for their constructive evaluation of our work. We are encouraged by the recognition of DDOME’s novelty in decentralized setting. We are also pleased the acknowledgment of the theoretical contributions and anchor-client mechanism. We also appreciate the attention to detail in identifying typos, which we will address in the final version. We respond to the main concerns raised.
>
>   1. **"Ambiguity in M=K Scenario: In experiments where the number of experts equals the number of selected experts (M=K), particularly for M=K=2 on CIFAR10, DDOME's performance (60.16\%) is lower than the initial common expert's accuracy (73.39\%) [Table 7]. The authors attribute this to "malicious 'interference'", which is a vague explanation. This suggests a potential fragility or a specific failure mode in DDOME under certain configurations that needs deeper investigation and mitigation."**
>      - _We believe the concern raised here is mostly due to a misunderstanding of the purpose of the experiment._ This is indeed a corner case. While the exact mechanism behind the drop in performance remains an open question, we believe this behavior is not entirely surprising. In the $M=K=2$ setting, all experts are sent to all active clients, which removes the intended benefit of selective routing. With a low number of experts, specialization becomes difficult as interference between updates is more likely.
>     **_But we would also like to clarify the purpose of this ablation experiment: it was designed to test DDOME under the same expert-sharing protocol as FedMix, where all experts are sent to each client. Even in this setup, our lightweight server aggregation and inference protocol still outperform FedMix._** So while performance does degrade relative to the initial expert, the experiment shows that DDOME still offers advantages even under these edge cases.
>
>    2. **"Small and Uninteresting datasets: Datasets considered from FL literature are too old and toyish from before 2021, and appear to be non-interesting. The authors should consider more interesting and larger datasets."**
>
>       -  We respectfully disagree with the premise of this statement. The datasets are far from toyish or small. While CIFAR10/100 and EMNIST may be considered outdated by some in the broader community, **_we emphasis that we are using their federated learning versions_**. These federated versions introduce significant non-i.i.d. data skew across clients, making them much more challenging. Federated CIFAR10/100 in particular continue to serve as standard and difficult benchmarks across numerous FL studies. Additionally,  Additionally, the Yahoo! dataset is one of the largest publicly available text classification datasets. Even our subsampled version demands several days of training, with some baselines taking up to a week to complete. DDOME itself requires several days of training. We also note that recent FL papers from NeurIPS 2024 continue to rely on the same datasets we use: for example, **A Bayesian Approach for Personalized Federated Learning in Heterogeneous Settings** (Makhija et al., 2024) uses CIFAR10/100 and MNIST; **FedAvP** (Hong et al., 2024) uses CIFAR10/100 and FEMNIST; and **FedGMKD** (Zhang et al., 2024) uses CIFAR10/100 and SVHN.
>
>   3. **"Heavy Dependence on Pretrained Common Expert: The success of DDOME is critically dependent on the availability and quality of an effective pretrained common expert. The ablation study explicitly shows that a poor common expert can lead to performance collapse. This is a significant practical limitation, as acquiring such a well-performing expert might be costly or not feasible in all decentralized scenarios."**
>
>      - We agree the common expert is critical to DDOME, as it informs routing decisions. As acknowledged in section 6, the method can indeed degrade when a high-quality expert is not available. That said, this reliance is not unique to DDOME—many FL and personalized learning methods depend on embedding mechanisms; **Federated Learning of Models Pre-Trained on Different Features with Consensus Graphs** (Ma et al., 2023), **FedPETuning** (Zhang et al., 2023). In practice, we believe the growing availability of high-quality pretrained models can help mitigate this limitation. Nonetheless, we acknowledge this as a key constraint and an important area for future work
>
>   4.  **"Choice of hyperparameters: The authors consider a different optimizer for DDOME than the one considered for baseline models in lines 241-243. This appears problematic without a solid justification. Perhaps the change in optimizer is the cause for some of the gains."**
>
>
>        - We highly thank the reviewer for bringing this up, as it gives us the opportunity to correct the record. The statement that DDOME’s main model used a different optimizer is inaccurate. All the main models for all methods were trained using the same SGDM optimizer to ensure a fair comparison. **A step decay learning rate (not exponential) was applied to the gating network**. This was done to stabilize the training of the routing policy. We would also like to clarify that router decay was limited to a single experiment—specifically, DDOME on the Yahoo! dataset as it presented additional challenges due to the complexity of its features. The correct text in the manuscript (Lines 241-243) should be:
>        "_For text classification, all client models are trained using SGDM with identical hyperparameters. All clients use a batch size of 16, and one local epoch per round. The gating function is trained using an SGD optimizer; for image classification, this uses a fixed learning rate of 0.001, while for text classification the same initial learning rate is used but it incorporates a step decay learning rate schedule_"
>
>    5. **"Relative Communication Cost: While DDOME reduces communication compared to some MoE methods like FedMix, the authors acknowledge it "still incurs higher communication overhead compared to traditional single-model methods". This is an important trade-off that should be more explicitly highlighted in the main claims regarding communication efficiency."**
>
>       - We agree and explicitly acknowledge this trade-off in our Limitations. All communication claims are explicitly made only in reference to FedMix, not traditional FL baselines. We’ve done our best to frame them accordingly.
>
>   6. **"Lack of Statistical Significance: The paper does not report error bars for its empirical results. This makes it challenging to ascertain the robustness and generalizability of the reported accuracy improvements across multiple runs and different random seeds."**
>
>       - We appreciate this point; another reviewer raised it as well. Several experiments include error metrics and were averaged over multiple runs (e.g., Fig. 5 and 8). Subfigures 8b and 8d correspond to Table 1. We omitted error bars in the main tables for consistency but included significance results in the appendix. That said, not all experiments include this level of statistical detail, particularly those involving larger-scale configurations. Many baselines from Table 2 require 3–7 days per run, depending on hardware availability. Repeating these experiments under current resource constraints would require several additional months of compute. Nonetheless We aim to include variance estimates in the appendix for EMNIST in the final version.
>
> **Questions**
>
>  _Q1._  **"Robustness to Common Expert Quality"**
>
>   - Thank you for this question, as it touches on a key aspect of DDOME’s deployability in varied environments. We address each part below
>
>      - **"(1) Adaptive or periodic updates to the common expert?:"**
>
>         This introduces challenges. The common expert is meant to be static and shared once. Updating it periodically would require re-sharing, effectively becoming a central router. While viable, this introduces synchronization and consistency issues—an interesting future direction.
>
>       - **"(2) Unavailable or sub-optimal common-expert?"**
>
>         There are several possible strategies:
>
>           - Leverage the rich transfer learning literature using pretrained models as plug-and-play components or with minimal fine-tuning.
>
>          - Explore lightweight data embedding or preprocessing techniques. An interesting direction that drops the common-expert entirely.
>
>         - Draw inspiration from the learning-to-optimize literature, by training a models solve the optimization based topk routing problem. Further fine-tuning on the actual federated task is possible.
>
>         -  A simple strategy is to increase the capacity of the router itself. A more expressive router might help mitigate the negative effects of sub-optimal embeddings.
>
>
>  _Q2._ **"Scalability to Million-Client Scale"**
>
>  - This is a very thoughtful question and sparked cool discussion in our group. At such scales rethinking the server–client protocol for any FL system is necessary. Communication and synchronization become key bottlenecks. In the context of DDOME, the gating function is the component most directly implicated. While it would certainly need to be adapted, we do not believe it needs to scale into a massive monolith to remain effective. One promising direction we see is client clustering. Clients are grouped based on various criteria, and routing is organized hierarchically:
>
>    -  At the first level, run the gating network once per cluster to select a small subset of experts.
>
>    -   At the second level, clients within each cluster:
>         -   Use local lightweight routers to select from the expert subset, or
>         -   Delegate routing and coordination to a cluster-specific intermediary server
>
>      To reduce global communication further we can leverage peer-to-peer (P2P) communication within clusters, and cache frequently used (“hot”) experts locally. Furthermore, we can apply asynchronous communication between clusters to avoid global synchronization bottlenecks.

---

> ### Author Response · Authors · 2025-08-06
> **Rebuttal discussion followup**
>
> We’d like to follow up on our rebuttal. With about two days left in the discussion period, we would appreciate any additional questions or feedback so we can address them promptly and engage in a constructive discussion.
>
> Thank you for your time and consideration.

---

### Official Review · Reviewer_pNK2 · 2025-07-13

**Clarity:** 4
**Significance:** 4
**Originality:** 3
**Rating:** 5
**Confidence:** 3

**Summary:**

Paper investigates how joint training of gating functions and experts in MoE frameworks can enable adaptive specialization in decentralized settings, especially in those without explicit task partitions. From what I understand the method uses data heterogenity across clients to achieve expert specialization.

**Questions:**

1. How does your method improve privacy preservation in the current federated learning scinarios. Detailed analysis would strengthen the paper
2. Is K in Top-K experts a hyperparameter? Is ther anyway K could be better estimated using the number of experts, nature of the task and other additional metrics?

**Ethical Concerns:**

["NO or VERY MINOR ethics concerns only"]

**Final Justification:**

Although this paper has some flaws, I think the contributions have significant scope and would help future work build better MoE systems. Besides, this paper is based on strong foundations as seen in the proofs from the Appendices. Due to this reason, I'm increasing my score to "accept"

**Limitations:**

Yes.

**Quality:**

3

**Strengths And Weaknesses:**

Strengths
1. paper is well written with formal proofs using theorem 1, which proves that sequential/disjoing training of gating functions and experts leads to suboptimal specialization
2. Elaborate ablation studies are performed, in situations involving increasing the clients pool, dynamically switching the client's pool
3. The section on Broader impacts and limitations outlines various limitations of the proposed method, that help in caping the expectations of the proposed method


weakness
1. Although the experiments show that the proposed method performs better than the baselines, I think the authors need to include more recent baselines. the most recent baseline that I see on the comparision table is from 2023.
2. As someone who worked in a related field but not the same research area, it'd be helpful if the authors can have a figure on an example of what are tasks and domains. This would help appreciate the contributions of the paper better
3. It'd be helpful to have one medicore expert architectures along with ResNet-34 and TinyBERT for text to understand how the perfrance of the method various with the change of expert architecture
4. It is slightly surprising to find the paper lacking error bars or stastical sgnificance testing across multiple runs, which make it difficult to asses the uncertainity and realiability of the perforamnce of the method

---

> ### Author Rebuttal · Authors · 2025-07-30
>
> We are encouraged that the reviewer found our paper to be well written and clear. We also appreciated the acknowledgment of the significance of the formal proof (Theorem 1) demonstrating the suboptimality of disjoint training of gating functions and experts, as this is something we wanted to highlight in our paper. We are pleased that the reviewer valued our elaborate ablation studies. We address some of the concerns and weaknesses raised by the reviewer below.
>
>   1. **"Although the experiments show that the proposed method performs better than the baselines, I think the authors need to include more recent baselines. the most recent baseline that I see on the comparision table is from 2023."**
>      - We would like to note that a similar concern was raised by one other reviewer, and we appreciate that multiple reviewers highlighted the importance of broader comparisons. We fully agree that including more baselines is valuable for situating our contribution within the current landscape. **That said, we would like to clarify the scope, intent, and novelty of our work. A central goal of our study is to provide a proof-of-concept for how distinct expert models—selected dynamically through a lightweight router—can lead to client-specialized training within a single-domain.** This setup is highly relevant for decentralized training, and through this investigation, we developed a state-of-the-art decentralized MoE framework that is competitive with strong FL baselines. While we agree that additional baselines could further reinforce the empirical analysis, we believe the current experimental results are sufficient to demonstrate the promise of our approach and lead to meaningful conclusions.
>
>    2. **"As someone who worked in a related field but not the same research area, it'd be helpful if the authors can have a figure on an example of what are tasks and domains. This would help appreciate the contributions of the paper better"**
>
>       - Thank you for the opportunity to clarify this. We did our best to provide some real-life examples of the setting in the introduction, but we understand that the terms "task" and "domain" can carry different meanings depending on the research context, and we appreciate the different perspective. In our work, we primarily focus on single-domain federated learning, where the data across clients may differ in distributional characteristics but generally share the same task. For example, consider pretraining a model that predicts the type and stage of cancer using CT scans. Here, cancer prediction is the task, and the clients participating in this training process are different hospitals or clinics with private medical data. Each client has labeled CT scans, but likely follows its own underlying distribution of the same task (e.g., one client might specialize in prostate cancer, while another specializes in breast cancer). In the final version, we will include better-contextualized explanation in the introduction (Lines 39 - 44)
>
>    3. **"It'd be helpful to have one medicore expert architectures along with ResNet-34 and TinyBERT for text to understand how the perfrance of the method various with the change of expert architecture"**
>
>       - We agree, this is something the group has been very interested in studying. Diverse architectures across the server has been something we want to study to better understand what can be achieved in such settings. The impact of varying expert architectures is indeed an important and interesting direction, but also a very big one. We have left a broader investigation of this question to future work.
>
>    4. **"It is slightly surprising to find the paper lacking error bars or stastical sgnificance testing across multiple runs, which make it difficult to asses the uncertainity and realiability of the perforamnce of the method"**
>
>        - We thank the reviewer for raising this important point. We fully agree that statistical robustness is essential for assessing the reliability of experimental results. **_As shown in Figure 5 and Figure 8, several of our experiments do include statistical error metrics and are averaged over multiple runs. In particular, subfigures (b) and (d) in Figure 8 correspond to results reported in Table 1_**. We omitted error bars in the main tables for consistency but included statistical significance results in the appendix. That said, we acknowledge that not all experiments include this level of statistical detail, especially those involving larger-scale setups. For instance, many of the configurations in Table 2 require 3–7 days per run, depending on available hardware. Running these experiments multiple times under our current resource constraints would require several additional months of compute. We appreciate the reviewer’s emphasis on this aspect, and we will aim to address it for the EMNIST dataset in the final version.
>
> Questions:
>
>    Q1. **"How does your method improve privacy preservation in the current federated learning scinarios. Detailed analysis would strengthen the paper"**
>
>    - To clarify any confusion, our paper does not claim to introduce new privacy-preserving mechanisms in the formal sense. Rather, we emphasize that our method operates fully within the standard privacy-preserving framework of federated learning. Within this framework, our method goes a step further by enabling zero-shot personalization without requiring any additional data or fine-tuning from new clients. While some FL approaches also support this property, many require further adaptation, data, or finetuning, which can introduce additional privacy risks. Our design avoids this, reducing client burden while maintaining personalization.
>
>  Q2. **"Is K in Top-K experts a hyperparameter? Is ther anyway K could be better estimated using the number of experts, nature of the task and other additional metrics?"**
>    - Yes, to the best of our knowledge,
> K is treated as a hyperparameter in nearly all existing MoE-based work, including ours. However, it is not arbitrary, choosing
> K involves balancing computational efficiency, communication cost, and model performance. While some empirical studies explore the effect of different
> K values, we are not aware of a principled or theoretical framework for setting
> K. Developing such a framework by tying
> K to the number of experts, task complexity, or client heterogeneity is a rich and largely unexplored direction that could support an entire line of future research. Thank you for this very interesting question!

---

> ### Author Response · Authors · 2025-08-06
> **Rebuttal discussion followup**
>
> We’d like to follow up on our rebuttal. With about two days left in the discussion period, we would appreciate any additional questions or feedback so we can address them promptly and engage in a constructive discussion.
>
> Thank you for your time and consideration.

---

> > ### Comment · Reviewer_pNK2 · 2025-08-09
> >
> > First of all, I thank the authors for answering every question and clarrifying the points that I had raised as "weakness". From what I had read from the rebuttal, I was better able to understand the novelty and scope of this work, compared to the current existing baselines. Furthermore, the questions that I raised do not target the novelty of the paper, but are helpful to study the scope of this paper.
> >
> > Although this paper has some flaws, I think the contributions have significant scope and would help future work build better MoE systems. Besides, this paper is based on strong foundations as seen in the proofs from the Appendices. Due to this reason, I'm increasing my score to "accept"

---

### Official Review · Reviewer_7ym6 · 2025-07-19

**Clarity:** 3
**Significance:** 2
**Originality:** 3
**Rating:** 4
**Confidence:** 3

**Summary:**

This paper investigates how the joint training of gating functions and experts can dynamically allocate domain-specific expertise across multiple underlying data distributions. Based on this insight, the authors propose the DDOME (Dynamically Decentralized Orchestration of MoEs) framework. Experimental results demonstrate noticeable performance improvements.

**Questions:**

- Include more recent methods in the experimental comparisons and explicitly evaluate communication cost as part of the comparison.

**Ethical Concerns:**

["NO or VERY MINOR ethics concerns only"]

**Final Justification:**

My concerns have been well addressed and I will raise my score.

**Limitations:**

Please refer to the weaknesses section above.

**Paper Formatting Concerns:**

No major formatting issues noticed.

**Quality:**

2

**Strengths And Weaknesses:**

Strengths
- Applying Mixture-of-Experts (MoE) to federated learning is a relatively novel perspective that opens up a range of interesting research questions.
- The paper is clearly written, and both the methodology and experimental results are presented in an intuitive manner.

Weaknesses
- Although communication cost is listed as a contribution, it is not thoroughly evaluated or compared in the experiments.
- The experimental section lacks comparisons with more recent federated learning methods that incorporate MoE, and additional baselines should be included.
- DDOME appears to benefit from using a larger number of experts (M), which may give it an unfair advantage in comparisons.

---

> ### Author Rebuttal · Authors · 2025-07-30
>
> We thank the reviewer for their feedback. We are encouraged that they found our application of MoE to federated learning to be a novel and promising direction, and that they appreciated the clarity of our writing and the intuitive presentation our methodology and experimental results. We are also pleased that the reviewer acknowledged DDOME as a technically sound framework that offers meaningful performance improvements across federated domains. We address the reviewer’s concerns below:
>
> 1. **"Although communication cost is listed as a contribution, it is not thoroughly evaluated or compared in the experiments."**
>
>       - We acknowledge that our current draft does not include an experimental evaluation of communication cost. **_However, we would like to clarify that our communication-related claim was specifically made in reference to FedMix,_** which requires all expert transmissions per client per round. In contrast, DDOME limits communication by decentralizing expert selection via the router. **_To back up our claim, we derive the communication cost of DDOME and FedMix in terms of the of parameters being transmitted_**. The cost is a function of the number of clients, communication rounds, and number of experts. We show via a simple analysis that FedMix's communication cost is bounded below by DDOMES communication cost. Proving that DDOME is indeed more efficient across all metrics of interest. Please find the analysis at the end
>
>
>   2. **"DDOME appears to benefit from using a larger number of experts (M), which may give it an unfair advantage in comparisons"**
>
>         - We believe that the reviewer might have some confusion about the conclusions from our experimental results. **_In all of our experiments, the number of experts communicated to and processed by each client is identical between DDOME and FedMix (Please refer to table 1, 2, and 3)_**. In fact, one of DDOME's contributions is its efficiency in managing a larger server-side expert pool without increasing this client-side cost or communication (Please refer to our communication cost analysis provided).  One of the research thrusts behind this work, and MoEs more broadly, is efficient and intelligent scaling.
>
>    3. **"The experimental section lacks comparisons with more recent federated learning methods that incorporate MoE, and additional baselines should be included."**
>
>       - We thank the reviewer for this suggestion. We agree that comparing against more related methods is essential for situating our contribution, and we would like to clarify the scope, intent, and novelty of our work to address this concern thoroughly. **_A central goal of our work is to provide a proof-of-concept for how distinct expert models — selected dynamically through a lightweight router — can lead to client-specialized training in a single domain_**.  This setting has proven itself to very relevant in decentralized training, specifically FL. As an outcome of this study we developed a state of the art decentralized MoE framework that is competitive with our FL baselines. We agree that additional baselines might provide more support to the study, but we believe that the experimental results provided in the paper already lead to a meaningful conclusion.
>
>
>
> **Communication Cost Analysis**
>
> **_Variable Definitions_**
>
> $M$: number of expert models.
>
> $R$: number of communication rounds.
>
> $S$: number of clients.
>
> $N$: active clients.
>
> $k$: topk experts ($k \le M$).
>
> $P_r$: parameters of router.
>
> $P_e$: parameters of a single expert.
>
> $P_c$: parameters of common expert.
>
> $\epsilon$: size to communicate an expert index.
>
> **_Analysis of DDOME_**
>
> **Communication Cost Derivation**
> Total cost = initial setup + cumulative round cost.
>
> - **Initial Cost ($C_{init}$):**
> This one-time, server-to-client cost involves sending the common expert model to all clients $S$.
> \begin{equation}
> C_{\text{init}} = S \cdot P_c
> \end{equation}
>
> **Per-Round Cost ($C_{round}$):**
> For each of the $R$ rounds, the cost is the sum of downlink (Server-to-Client) and uplink communication (Client-to-Server).
>
> - **Downlink Cost ($C_{down}$):** Router ($P_r$) and $k$ experts to  $N$ active clients.
> \begin{equation}
> C_{down} = N(P_r + k \cdot P_e)
> \end{equation}
>
> - **Uplink Cost ($C_{up}$):**  top-$k$ indices + updates for router + $k$ experts to $N$ active clients.
> \begin{equation}
> C_{up} = N (k \cdot \epsilon + P_r + k \cdot P_e)
> \end{equation}
>
> **Note:** The communication here does not represent computational order.
>
> The total cost for a single round is therefore:
> \begin{align*}
> C_{\text{round}} = C_{\text{down}} + C_{\text{up}} \\
> = N (2 P_r + 2 k P_e + k \epsilon)
> \end{align*}
>
> **Total Communication Cost ($C_{total}$)**
> The total cost over $R$ rounds is the sum of the initial cost and the cumulative round costs.
> \begin{align*}
> C_{total} &= C_{init} + R \cdot C_{round} \\
> = (S \cdot P_c) + R \cdot N (2 P_r + 2 k P_e + k \epsilon)
> \end{align*}
>
> Since the parameter sizes are much larger than index sizes ($P_r, P_e \gg \epsilon$), we can approximate the total cost as:
> \begin{equation}
> C_{\text{total}} \approx (S \cdot P_c) + 2 R \cdot N (P_r + k P_e)
> \end{equation}
>
> **_Analysis of FedMix_**
>
> The FedMix algorithm communicates all experts and the router in every round.
> **Communication Cost Derivation**
>
> **Per-Round Cost ($C_{\text{round, FedMix}}$)**
> In each round, every active client downloads and uploads all the models.
>
> - **Downlink Cost ($C_{\text{down, FedMix}}$):** The server sends all $M$ experts ($M \cdot P_e$) to each of the $N$ active clients.
> \begin{equation}
> C_{\text{down, FedMix}} = N (M \cdot P_e)
> \end{equation}
>
> - **Uplink Cost ($C_{\text{up, FedMix}}$):** Each of the $N_{\text{active}}$ clients sends back the updated parameters of the $M$ experts.
> \begin{equation}
> C_{\text{up, FedMix}} = N (M \cdot P_e)
> \end{equation}
>
> **Total Communication Cost ($C_{\text{total, FedMix}}$)**
> The total cost is the per-round cost multiplied by the number of rounds, $R$.
> \begin{equation}
> C_{\text{total, FedMix}} = 2 R \cdot N ( M \cdot P_e)
> \end{equation}
>
> **_Limit Analysis_**
> This analysis examines the asymptotic behavior of the communication costs under the following assumptions:
> . The number of active clients is a fixed fraction of the total: $N = S/c$ for some constant $c > 0$.
> . The parameter size of the initial common expert is equal to that of a single expert: $P_c = P_e$.  **Note:** This is true in our experiments.
>
> - **Analysis as the number of experts $M \to \infty$**
>
>   - **DDOME**
>   \begin{align*}
>   \lim_{M \to \infty} C_{\text{total}} &= \lim_{M \to \infty} \left[ (S \cdot P_c) + 2 R \cdot N (P_r + k P_e) \right] \\
>   = (S \cdot P_c) + 2 R \cdot N (P_r + k P_e)
>   \end{align*}
>   The cost is independent of $M$, implying that the communication cost is bounded and does not increase as the number of experts grow.
>
>   - **FedMix**
>   \begin{align*}
>   \lim_{M \to \infty} C_{\text{total, FedMix}} &= \lim_{M \to \infty} \left[ 2 R \cdot N ( M \cdot P_e) \right] \\
>   = \infty
>   \end{align*}
>
>  The cost is a linear function of $M$. As the number of experts grows, the cost grows without bound. This means the communication cost is unbounded, making the algorithm very unscalable compared to DDOME.
>
>
> - **Analysis as Number of Rounds $R \to \infty$**
> We analyze the cost as the training process becomes infinitely long.
>
>   - **DDOME**
>   The cost is of course unbounded but grows linearly with $R$. The rate of growth is given by the partial derivative with respect to $R$:
>   \begin{equation}
>   \frac{\partial C_{\text{total}}}{\partial R} = \frac{2S}{c} (P_r + k P_e)
>   \end{equation}
>
>   - **FedMix**
>   The rate of growth with respect to $R$ is:
>   \begin{equation}
>   \frac{\partial C_{\text{total, FedMix}}}{\partial R} = \frac{2S}{c} (M P_e)
>   \end{equation}
>
> **Comparison**
> Both costs are unbounded as $R \to \infty$. However, comparing their growth rates reveals that for $k \ll M$, DDOME grows significantly slower, making it more efficient for longer training durations.
>
> $\frac{\frac{2S}{c} (P_r + k P_e)}{\frac{2S}{c} (M P_e)} < 1$
>
> - **Analysis as Number of Clients $S \to \infty$**
> In this case we will add the $\epsilon$ term back in, as it directly depends on $S$.
>
>   - **DDOME**
>   The rate of growth with respect to $S$ is:
>   \begin{equation}
>   \frac{\partial C_{\text{total}}}{\partial S} = P_e + \frac{R}{c} \left(2P_r + 2kP_e + k \epsilon \right)
>   \end{equation}
>
>   - **FedMix Algorithm**
>   The rate of growth with respect to $S$ is:
>   \begin{equation}
>   \frac{\partial C_{\text{total, FedMix}}}{\partial S} = \frac{2R}{c} ( M P_e)
>   \end{equation}
>
> **Comparison**
> The goal is to prove that FedMix  rate of change is greater. We will show that this is indeed the case under certain (realistic) conditions.
>
> We seek to demonstrate that:
> $2\frac{R}{c} M P_e > P_e + \frac{R}{c} (2P_r + 2 k P_e + k \epsilon)$
>
> We proceed by rearranging this inequality to find the condition under which it holds true.
>
> $\frac{R}{c} \left[ 2(M - k) - \frac{2P_r}{P_e} - \frac{k \epsilon}{P_e} \right] > 1$
>
> $2(M - k) - \frac{2P_r}{P_e} - \frac{k \epsilon}{P_e}$, is a positive number slightly less than $2(M-k)$ and almost certainly greater than 1.
>
> Thus, the inequality FedMix $>$ DDOME holds true if:
>
> $\mathbf{R > \frac{c}{2(M - k) - \frac{2P_r}{P_e} - \frac{k \epsilon}{P_e}}}$
>
> This condition is generally satisfied in practical scenario. For example we can plug in our own experimental values from cifar10 and Yahoo!.
>
> $M = 5$, $k=2$, $c = 10$, $R = 2000$  $P_e \gg P_r$, $P_e \gg \epsilon$.
>
> In this case it is easy to see that this inequality is indeed satisfied, showing us that the rate of change of the communication cost of FedMix, as a function of the number of clients will remain bounded below by DDOME.
>
> **_Conclusion_**
>
> Across all scaling dimensions of interest—experts (M), rounds (R), and clients (S)—DDOME communication is fundamentally more efficient and scalable.

---

> ### Author Response · Authors · 2025-08-06
> **Rebuttal discussion followup**
>
> We’d like to follow up on our rebuttal. With about two days left in the discussion period, we would appreciate any additional questions or feedback so we can address them promptly and engage in a constructive discussion.
>
> Thank you for your time and consideration.

---

### Note · Authors · 2025-08-15

Dear Program Chairs, Senior Area Chairs, and Area Chairs,

We sincerely thank the reviewers for their invaluable and detailed feedback. We are glad that the reviewers find our paper well written (7ym6, pNK2, tKQk), our methodology clear and significant (7ym6, pNK2, Uwk3), and our theoretical result solid (pNK2, Uwk3, tKQk). We are grateful that our rebuttals helped clarify key aspects of our work, leading two reviewers (pNK2, Uwk3) to raise their scores in support of our paper.

We particularly appreciate Reviewer tKQk’s comment regarding our commitment to revisions. We apologize if our rebuttal seemed dismissive; that was not our intent. We are fully committed to incorporating the excellent feedback into our final manuscript. Specifically, we will enhance the introduction to clarify that our theoretical insights and the DDOME framework are applicable to general MoE settings, with Federated Learning serving as a critical testbed. We will also revise the notation and formatting for clarity, including removing the stylistic quotes and renaming the “common expert” to “common embedder” to avoid confusion.

The discussion period was instrumental in clarifying several points we wish to re-emphasize for the committee’s consideration. First, our communication cost analysis, provided in the rebuttal, theoretically confirms DDOME’s superior scalability over other MoE-based FL methods. Second, we clarified a misunderstanding regarding experimental fairness (Reviewer Uwk3): all core models were trained with the same optimizer, with a minor, justified modification to the gating network’s scheduler in only one experiment to ensure stability.
Finally, we want to highlight that our novel anchor-client mechanism is not just a workaround but a principled alternative to standard load-balancing, designed specifically to foster stable specialization in decentralized settings; a key contribution of our work.
We are confident that these revisions will strengthen the paper and are grateful for the opportunity to present our work to the NeurIPS community.

Sincerely,

The Authors of Submission #22435

---

### Decision · Program_Chairs · 2025-09-17

**Decision:**

Accept (poster)

**Comment:**

This paper studies expert specialization in MOE setting in context of Federated Learning. Using theoretical insights from their framework, authors argue that joint training of experts and gating weights is necessary to make sure training loss continues to decrease. Authors additionally propose dynamic expert allocation framework to achieve speciliazation without the explicit need for labels. Experiments on both image and text classification tasks and thorough ablations are convincing and achieve SOTA performance.

All the reviewers find the setting of MOE in FL challenging, impactful and interesting. Additionally the paper is well written, easy to follow, experiments are comprehensive and theoretical insights are used to propose new methodology. While reviewers outlined several problems with an experimental setup, authors did a great job during rebuttal phase, with several reviewers increasing the score due to author's clarifications. We expect that all the additional experiments and clarifications are incorporated in the final draft.